# Influence of Precise Products on the Day-Boundary Discontinuities in GNSS Carrier Phase Time Transfer

**DOI:** 10.3390/s21041156

**Published:** 2021-02-06

**Authors:** Xiangbo Zhang, Ji Guo, Yonghui Hu, Baoqi Sun, Jianfeng Wu, Dangli Zhao, Zaimin He

**Affiliations:** 1National Time Service Center, Chinese Academy of Sciences, Xi’an 710600, China; guoji@ntsc.ac.cn (J.G.); huyh@ntsc.ac.cn (Y.H.); sunbaoqi@ntsc.ac.cn (B.S.); wujianf@ntsc.ac.cn (J.W.); zdli@ntsc.ac.cn (D.Z.); hezm@ntsc.ac.cn (Z.H.); 2University of Chinese Academy of Sciences, Beijing 100049, China; 3Key Laboratory of Time and Frequency Primary Standards, Chinese Academy of Sciences, Xi’an 710600, China; 4Key Laboratory of Precise Positioning and Timing Technology, Chinese Academy of Sciences, Xi’an 710600, China

**Keywords:** GNSS, precise point positioning, time comparison, station clock estimates, day-boundary discontinuity, precise products, frequency stability

## Abstract

Global navigation satellite system (GNSS) precise point positioning (PPP) has been widely used for high-precision time and frequency transfer. However, the day-boundary discontinuities at the boundary epochs of adjacent days or batches are the most significant obstacle preventing PPP from continuous time transfer. The day-boundary discontinuities in station estimates and time comparisons are mainly caused by the code-pseudorange noise during the analysis of observation data in daily batches, where the absolute clock offset is determined by the average code measurements. However, some discontinuities with amplitudes even more than 0.15 ns may still appear in station clock estimates and time comparisons, although several methods had been proposed to remove such discontinuities. The residual small amplitude of the day-boundary discontinuities in some PPP station clock estimates and time comparisons through new GNSSs like Galileo seems larger, especially using precise clock products with large discontinuities. To further understand the origin of the day-boundary discontinuities, the influence of GNSS precise products on the day-boundary discontinuities in PPP station clock estimates and time comparisons is investigated in this paper. Ten whole days of Multi-GNSS Experiment (MGEX) from modified Julian date (MJD) 59028 to 59037 are used as the observation data. For a comparative analysis, the station clock estimates are compared with global positioning system (GPS) and Galileo observations through PPP and network solutions, separately. The experimental results show that the daily discontinuities in current combined GPS final and rapid clock products are less than 0.1 ns, and their influence on the origin of day-boundary discontinuities in PPP station clock estimates and time comparison are statistically negligible. However, the daily discontinuities in individual Analysis Centers (ACs) GPS products are more extensive, and their influence on the origin of the day-boundary discontinuities in GPS PPP station clock estimates cannot be ignored. The day-boundary discontinuities demonstrate random walk noise characteristics and deteriorate the station clocks’ long-term frequency stability, especially at an average time of more than one day. Although Galileo clock daily discontinuities are different from those of GPS, their influence on the day-boundary discontinuities in station clock estimates is nearly similar to the GPS PPP. The influence of daily discontinuities of Galileo clocks on PPP time comparison is similar to GPS and is not particularly critical to time comparison. However, combined and weighted MGEX products should be developed or Galileo IPPP should be used for remote comparison of high-stability clocks.

## 1. Introduction

Since the geodetic time transfer concept was proposed in 1990, global navigation satellite system (GNSS), based on carrier-phase measurements, has been recognized and demonstrated as a high-accuracy method for time and frequency transfer [1]. The jointly sponsored International GNSS Service (IGS)/Bureau International des Poids et Mesures (BIPM) Pilot Project to Study Accurate Time and Frequency Comparisons using GPS Phase and Code Measurements began in 1998. This pilot project aimed to develop strategies for improving the accurate global time and frequency comparisons using geodetic global positioning system (GPS) techniques [2,3]. The project mutually benefits both IGS and BIPM because it not only helps the IGS to refine its clock products and link them to universal time coordinated (UTC), but also can provide new time transfer results for the BIPM to improve the formation of international atomic time (TAI) [4,5]. As one of the most commonly used techniques for GNSS geodetic time transfer with the precise orbit and clock products provided by the IGS, GPS precise point positioning (PPP) has been formally recommended by BIPM for the computation of TAI and UTC since September 2009 [6,7]. Over 50% of more than 70 national timing institutions have currently adjusted to such time links, using them for remote clock comparison. GPS PPP precision can be reach about 100 ps, which is more precise than the traditional method of common-view (CV) and less expensive than two-way satellite time and frequency transfer (TWSTFT). PPP provides not only superior short-term stability due to the low noise but also good long-term stability and higher resolution. However, the PPP cannot be considered a continuous time transfer method; its long-term frequency stability is affected due to the day-boundary discontinuities causing time jumps at the boundary epochs of two adjacent batches [8]. In some stations, the day-boundary discontinuities may reach more than 1 ns, thereby restrict PPP’s use in high-precision applications, like frequency comparisons of primary and secondary standards [9,10]. 

In the past 15 years, various studies have been devoted to the origin of day-boundary discontinuities [11,12]. Color code-pseudorange noise is generally responsible for such discontinuities in PPP time transfer with daily batches due to the dependence of the absolute clock estimates obtained from the code-pseudorange measurements [13]. To achieve continuous time transfer, various rigorous approaches have been proposed to reduce the day-boundary discontinuities, including shift and overlapping, ambiguity stacking, longer and sliding batches, phase-only, and rinex-shift [14,15,16,17,18,19]. Although these methods provide a continuous time transfer solution, their long-term clock stability comparison are confined by some errors and new noises. A small amplitude of discontinuities, even more than 0.15ns, can still be observed at the day boundaries of PPP time transfer. This demonstrates that code-pseudorange measurement noise is responsible for the large amplitude of the day-boundary discontinuities; other underlying causes, like nonperfect of IGS orbit and clock files at midnight, are also responsible. The integer-PPP (IPPP) technique, proposed by the Centre National d’Etudes Spatiales (CNES) and Collecte Localisation Satellites (CLS) based on the precise products of GPS wide-lane satellite bias (WSB), is useful for achieving continuous time and frequency transfer [20,21]. Compared with the classical PPP, this method can improve the stability of frequency comparisons at an averaging time of a few hours and above, significantly more than 5 × 10^−16^ at a 1 day averaging time [22]. However, it highly relies on satellite phase bias products. Since phase bias products are limited for new GNSS, such as the Beidou Satellite System (BDS), Galileo, and QZSS. IPPP has been usually adopted for international time and frequency transfer only through GPS. In the near future, Galileo IPPP will hopefully also be used for remote clock comparison.

PPP highly depends on external precise satellite orbit and clock products, and all products provided by IGS analysis centers (ACs) are based on GNSS observations. Similar to PPP, the measurement noises in observations also affect the estimation of orbit and clock parameters. Different processing strategies have been adopted by all ACs for satellite orbit and clock generation, so the quality of precise products provided by different ACs differs. To improve the accuracy and robustness of GPS products, IGS combined and weighted products submitted by several ACs to generate its final and rapid products. To meet the timing project demand, IGS and BIPM jointly established the final IGS time scale (IGST) and rapid IGS time scale (IGRT) as respective time references for IGSFinal and IGSRapid products, which are more continuous than the fixed station clock considered as the time reference in most individual ACs products. However, discontinuities remain at the boundaries of PPP station clock estimates with IGS combined products. To satisfy the new constellations like Galileo, BDS, and QZSS, IGS initiated the Multi-GNSS Experiment (MGEX) pilot project in 2011 to generate precise mixed orbit and clock products for all available GNSSs. As GPS PPP has been successfully used for national time and frequency comparisons from its early days, the studies performed on PPP time transfer through other GNSSs like Galileo and BDS reported promising results [23,24]. Although several individual ACs have provided products that are compatible with GPS, there have been no combined and weighted MGEX products, until now, especially for Galileo. Different individual ACs provide orbit and clock products with different qualities. Therefore, the discontinuities at the midnight boundary from day to day in Galileo orbit and clock products employed for time and frequency transfer can affect the PPP and induce new errors in station clock estimates. A network processing solution with batch least square was proposed to eliminate the day-boundary discontinuities and avoid using external precise clock products [25]. Although it is very similar to PPP, which employs combined code and carrier phase measurements, it is not affected by external precise clock products. Thus, classical PPP and the network solution should be compared to understand the contribution of discontinuities in GPS and Galileo precise products in the origin of day-boundary discontinuities in station clock estimates and time comparison.

In this study, we investigated the effect of discontinuities in GNSS precise products on PPP station clock estimates and time comparisons. In Section 2, the classical PPP model and the proposed network processing solution are introduced. Background information on the current GPS and Galileo precise orbit and clock products provided by several typical IGS and MGEX ACs are presented, and the daily discontinuities in their GPS and Galileo precise clock products are characterized through statistical analysis. Section 3 presents the experimental design using six timing MGEX stations with GPS and Galileo observations and data processing strategies to evaluate the impact of discontinuities in precise products on PPP clock estimates and time comparison results. The analysis and discussion of experimental results are described in Section 4. Finally, our conclusions are drawn in Section 5.

## 2. Principle and Methods

### 2.1. The PPP Principle

The classical PPP is a geodetic single-station post-processing solution for recovering precise coordinates of antennas, station clock offsets, and other parameters. It is based on measurements of pseudo-range and phase ionosphere-free combinations. The observation equations of pseudo-range and phase can be described with the following two equations, respectively:(1)$$P_{r,j}^{S}=\rho_{r}^{S}+cdt_{r}+d_{trop}+c(B_{P_{{}_{j}}^{S}}+b_{P_{{}_{r,j}}^{S}})+\varepsilon_{P_{r,j}^{S}},$$
(2)$$\Phi_{r,j}^{S}=\rho_{r}^{S}+cdt_{r}+d_{trop}+c(B_{\Phi_{{}_{j}}^{S}}+b_{\Phi_{{}_{r,j}}^{S}})+\lambda_{j}^{S}N_{r,j}^{S}+\varepsilon_{\Phi_{r,j}^{S}},$$
where r, S, and *j* (*j* = 1, 2) represent the receiver, satellite, and carrier frequency band, respectively; $P_{r,j}^{S}$ and $\Phi_{r,j}^{S}$ are the combined ionosphere-free code-pseudorange and carrier-phase measurements (m) for GPS L1/L2 or Galileo E1/E5a, respectively; $\rho_{r}^{S}$ is the geometric distance between receiver and satellite (m); $dt_{r}$ donates the receiver clock offsets (s); c is the speed of light in a vacuum (m/s); $d_{trop}$ is the tropospheric delay (m); $b_{P_{{}_{r,j}}^{S}}$ and $B_{P_{{}_{j}}^{S}}$ denote the uncalibrated code delays for the receiver and satellite (s), respectively; $b_{\Phi_{{}_{r,j}}^{S}}$ and $B_{\Phi_{{}_{j}}^{S}}$ denote the uncalibrated phase delays for the receiver and satellite (s), respectively; $\varepsilon_{P_{r,j}^{S}}$ and $\varepsilon_{\Phi_{r,j}^{S}}$ indicate the measurement noise and unmodeled residual errors in the combined ionosphere-free code and carrier-phase observations (s), respectively; $N_{r,j}^{S}$ denotes the combined phase ambiguity; and $\lambda_{j}^{S}$ is the wavelength of carrier frequencies (m/cycle). Equations (1) and (2) should be linearized to estimate parameters including the receiver’s precise coordinates, station clock offset, tropospheric delays, and phase ambiguities using the batch least squares algorithm. After estimating the station clocks between two different places, time transfer can be inferred from subsequent subtraction.

Some particular points should be emphasized. First, to estimate the parameters, it is assumed that satellite orbit and clock errors are corrected based on IGS or MGEX precise orbit and clock products. The errors induced by satellite and receiver antenna phase center offsets (PCOs) and phase center variations (PCVs) are corrected using IGS products for GPS and estimated MGEX for Galileo. The corresponding errors, including relativistic effects, Sagnac effect, tidal loadings, and phase windup, are corrected using the mature models in our PPP algorithm. Secondly, the tropospheric delay only refers to wet delays because the hydrostatic part is always stable, but the wet part changes quickly. A proper weighting scheme should be adjusted for the codes and carrier phases corresponding to the noise level of each observation type and satellite elevation: it is 0.3 m/0.003 m for GPS and 0.22 m/0.004 m for Galileo, and the satellite elevation is 10 degrees. Thirdly, the phase ambiguity obtained in this paper is a non-integer term due to the absorption of satellite and receiver hardware delays in the code-pseudorange and carrier-phase measurements. Finally, although code division multiple access signals are adopted by both GPSs of Galileo, the inter-system bias (ISB) should be appropriately considered in the case of PPP time transfer through Galileo when using different products. According to [26], the ISBs are related to receiver-independent time differences in different GNSS satellite clock products, and a receiver-dependent bias maybe absorbed by receiver clock offset. However, the mechanism has not been clear until now.

### 2.2. GPS and Galileo Precise Products and the Discontinuities Problem in Clock Products

Currently, there are nearly ten IGS ACs that contribute daily GPS orbit and clock solutions to the IGS combinations [27,28], three of which are Center for Orbit Determination of Europe (CODE), GeoForschungsZentrumPotsdam (GFZ), and Centre National d’Etudes Spatiales and Collecte Localisation Satellites (CNES-CLS). For brevity, the orbit and clock products provided by the above three IGS ACs are denoted as CODFinal, GFZFinal, and GRGFinal, respectively. Since GPS precise orbit and clock products provided by the above ACs employ different processing strategies, some differences exist between them. IGS has been generating the combined final and rapid clock products to improve the quality of products for precise positioning and timing, denoted by IGSFinal and IGSRapid, respectively. Due to the most up-to-date models and analysis strategies, the best quality and performance can be attained using the IGSFinal orbit and clock products, as well as the internal consistency of all IGS products. The IGSFinal products can be obtained weekly with a delay of nearly up to 14 days. The quality of IGSRapid orbit and products is nearly comparable to that of the IGSFinal and is available daily with a delay of 17 h [29].

At present, seven MGEX ACs are providing mixed multi-GNSS orbit and clock products. The Galileo constellation, CODE, GFZ, and CNES-CLS, all process Galileo observations and provide multi-GNSS mixed orbit and clock products denoted as COD, GBM, and GRG, respectively, to distinguish them from individual GPS products. Although three MGEX ACs employ similar code and carrier-phase observations to form the ionosphere-free linear combination E1/E5a for Galileo, their processing strategies for generating Galileo products are different. Specifically, CODE adopts a 3-day-arc with double-differenced network processing of phase observations to estimate satellite orbits using more than 100 MGEX stations providing daily RINEX3 files from late 2019. The original model of solar radiation pressure (SRP), which was applied to all satellites, was called ECOM1 before 2015, but has been changed into the new model called ECOM2 [30]. The orbits and other geometry-related information, including ERPs, station coordinates, and troposphere delays, are all fixed when solving satellite and station clocks. The COD clock is based on the zero-difference network processing of the ionosphere-free linear combination of code and phase, while the processing standards and background models are compatible with the orbit solution [31,32,33]. All ACs directly employ undifferenced data to calculate Galileo satellite orbits, except for CODE [34,35,36]. CNES-CLS is only an MGEX AC, generating orbits and clocks directly in one step in daily intervals based on more than 60 MGEX stations with Galileo observations. However, it employs an additional 6 h of data from the neighboring days (6 h+ 24 h+ 6 h) to obtain smaller discontinuities at the day boundaries [37]. Conventional PCOs proposed for MGEX are employed as the PCOs by CNES-CLS for Galileo products, while CODE adopts estimated and nonstandard PCO/PVO values for Galileo, and GFZ employs the corrected MGEX values [38]. CNES-CLS adopts the direct solar pressure and albedo plus IR for SRP. CNES-CLS uses TurboEdit, which is based on undifferenced code and phase for clock solution, with epoch sizes of 30 and 300 s for satellite and station clocks, respectively. Since GPS week 2022, CNES-CLS began to employ the undifferenced ambiguities fixing to estimate Galileo products [39,40]. GFZ also employs a 72 h arc for orbit solution and undifferenced code and phase data for clock solution based on more than 100 MGEX stations. Unlike CODE and CNES-CLS, GFZ uses the penumbra model without the a priori model for SRP. GFZ’s rapid daily clock files are generated by undifferenced phase and code observations with sampling times of minutes for stations and 30 s for satellites. The integer phase ambiguities are fixed for Galileo clock products using the method proposed by Ge [41]. Finally, The reference times are different for the above three kinds of Galileo products rather than using a combined time scale like IGST or IGRT. CODE, GFZ, and CNES-CLS adopt different schemes to align the satellite clock estimates and achieve continuous clock estimates.

Although all ACs and IGS adopt special techniques to achieve continuous satellite clock estimates, discontinuities still emerge at the day boundaries of GNSS precise products, whether GPS-specific or mixed MGEX products. These discontinuities are usually used as a standard to evaluate the quality of satellite clock products [42]. Statistical comparison of the average absolute values of discontinuities in five GPS-specific clock products and three Galileo-specific clock products during 10 days (MJD 59029 to 59037) clearly illustrated the problem of discontinuities in different products. Due to the perfect processing strategies adopted by IGS and ACs, the combined GPS clock daily discontinuities are very small. Thus, the daily discontinuities in GPS clock products are calculated as the difference between the final epoch of the last day and the first epoch of the next day. For most Galileo clocks, the clock stabilities are excellent, so the daily discontinuities are between the average absolute value of the last 5 min of the last day’s epochs and the first 5 min of the next day’s epochs. Due to the GBM Galileo clocks’ problems during the study period, we did not consider GBM clock products. Figure 1 and Figure 2 describe the daily discontinuities in GPS-specific clocks and Galileo products, respectively. As shown in Figure 1, the daily discontinuities in GPS combined IGSFinal and IGSRapid products are all within 0.10 ns for most GPS clocks, except for G01,G03, G10, G11, G27, and G28. Such discontinuities in these GPS clocks should be ascribed to clock instability and solution noise. However, the daily discontinuities in CODFinal, GFZFinal, and GRGFinal products were previously larger, and their amplitudes are nearly the same: within 0.25 ns for most GPS clocks, except for G03, G07, G10, G11, G18, G27, and G29. For comparison, we also calculated the daily discontinuities in GPS clocks generated by our network solution. The network clocks are comparable to the above 3 ACs GPS clocks. The clock solution noise and instabilities may cause the large daily discontinuities in those clocks.

As shown in Figure 2, the level of discontinuities in the three Galileo clock products do not vary greatly. Overall, the discontinuities in GRG and network clocks are almost the same, being a little higher than those in COD clocks; their values are commonly within 0.25 ns for most Galileo clocks. However, this is not true for the daily discontinuities in special Galileo clocks, such as E11, E12, E14, E18, E24, E26, and E30. The value of the daily discontinuities in E11, E26, and E30 exceed 0.7, 1.2, and 0.9 ns, respectively. The differences in clock daily discontinuities in the three Galileo products may be due to different processing strategies; however, the employed Rubidium clock’s instability and solution noise seem to be ascribed to E11 discontinuities. For E24, E26, and E30, there are always large discontinuities, which should be ascribed to the problems with the clocks. Compared to Figure 1, the daily discontinuities in Galileo clocks seem to be slightly larger than those in GPS clocks, and the differences in the discontinuities between COD and GRG Galileo clocks are due to different strategies, solution noise, or the different models adopted for orbits and clocks. Since the clock products are usually generated after that orbit and firmly consistent with orbit products, the deficiencies in satellite orbits impact the clocks.

### 2.3. Network Processing Solution to Eliminate Day-Boundary Discontinuities in PPP

As mentioned above, PPP is a single-station solution and highly dependent on external precise clock products. The clock products with large daily discontinuities are employed in PPP, affecting station clock estimates and time comparisons. Network processing solutions, as a method for estimating station clocks with the batch least-squares algorithm, have been proposed to overcome the problem of relying on external clock products, vastly eliminating the day-boundary discontinuities [25]. Although it can process GNSS observations with a similar zero-difference principle as PPP with dual-frequency ionosphere-free combination, it requires a set of stations distributed worldwide. It can estimate the station clock offsets with respect to a reference clock, unlike PPP station clock estimates with respect to IGST or IGRT. The reference clock can be chosen as a station with a high-stability external time scale to ensure the estimated station clocks’ continuity. It can be implemented with a multi-day batch to eliminate the day-boundary discontinuities at the adjacent days within one batch. Once the interested station clock offsets are estimated, time comparison results can be obtained through subsequent subtraction epoch-by-epoch between two station clock estimates. Compared with daily PPP, the network solution does not rely on external precise clock products provided by IGS or MGEX ACs. The precise station coordinates from PPP can be used as input values, and the orbits are based on the orbits predicted by IGS or MGEX. Unlike PPP, several stations can be processed together in the network solution, and the station clock estimates of each station can benefit from the measurements of all stations, making it more robust and accurate than PPP in estimating station clocks. Although the precision obtained with the network solution is equal to that obtained with the classical PPP, the uncertainty in the ambiguities between daily batches is lower than PPP, which helps to reduce the day-boundary discontinuities and achieve a continuous time comparison.

## 3. Experimental Design and Data Processing

Several station clock estimates experiments were performed through PPP and network processing solutions to evaluate the influence of GNSS precise products on the day-boundary discontinuities in the PPP results. Sixty ground MGEX stations evenly distributed worldwide were chosen to form the network, and station BRUX at the Royal Observatory of Belgium (Brussels) steered by UTC(ROB) was chosen as the reference clock due to its high-stability clock. Six test stations with mixed multi-GNSS observations, including PTBB, BRUX, WAB2, SPT0, OP71, ROAG, and USN8, are not only MGEX sites but also national timing laboratories participating in the computation of TAI and UTC; their equipment configurations are shown in Table 1. The receivers in these stations are connected by external H-maser clocks and are steered by UTC(PTB), UTC(CH), UTC(SP), UTC(OP), UTC(ROA), and UTC(USNO), respectively. The observation period of 10 whole days from MJD 59028 to 59037 was processed under the same daily PPP algorithm with IGSFinal, IGSRapid, CODFinal, GRGFinal, and GFZFinal products for GPS, and COD and GRG products for Galileo. Then, the absolute values of day-boundary discontinuities in PPP station clock estimates were calculated. For an exact comparison, the statistical results of day-boundary discontinuities in the network processing solution are presented as a reference for demonstrating the differences between daily PPP and network solutions. The effects of day-boundary discontinuities on station clocks’ frequency stability were also evaluated by calculating the modified Allan deviation (MDEV) to prove our evaluation.

Afterward, three specific links, including SPT0-PTBB, ROAG-PTBB, and USN8-PTBB, were established, and time comparison experiments were accomplished through GPS-only and Galileo-only PPP to investigate the influence of the discontinuities in GNSS products on time comparisons, especially through new GNSSs. The lengths of the above three time links were chosen as 622, 2182 and 6270 km, respectively, representing intra-European continental and transatlantic links. The day-boundary discontinuities in the time comparison results were then calculated to perform a comparative analysis. The frequency stability of the three time links with different GPS and Galileo products were compared.

As abovementioned, ISBs may influence the day-boundary discontinuities in station clock estimates and time comparisons through PPP and network solutions, but the mechanism was not clear until now. Thus, the effect of ISBs was not considered in this study.

## 4. Results and Analysis

### 4.1. Influence of Discontinuities in Different GPS Products on PPP Station Clock Estimates

Figure 3, Figure 4, Figure 5, Figure 6, Figure 7 and Figure 8 describe the station clock estimates of PTBB, WAB2, SPT0, OP71, ROAG, and USN8 through GPS PPP with IGSFinal, IGSRapid, CODFinal, GFZFinal, and GRGFinal products, respectively. The network solution and the statistical results of day-boundary discontinuities in station clock estimates are also presented. A mean was removed from the time series of all station clocks, and a 0.5 ns offset was considered for the series for display purposes. There are other errors at the boundary epochs, especially with GRG products. They were induced by the PPP clock estimation convergence and moved out in the average absolute value of the day-boundary discontinuities.

Figure 3, Figure 4, Figure 5, Figure 6, Figure 7 and Figure 8 shows that day-boundary discontinuities in PPP station clock estimates do not change significantly as the GPS combines precise products, but they change significantly for individual ACs GPS products; the whole levels of discontinuities in all station clock estimates are all within 0.25 ns except for some special cases. However, the whole level of discontinuities in all PPP station clock estimates with IGSFinal and IGSRapid products is less than those with CODFinal, GRGFinal, and GFZFinal products, but there are still some differences. More noise can be seen in the time series of the station clock with IGSRapid products.

For comparison, the level of day-boundary discontinuities in station clock estimates through the network solution is roughly less than those in the PPP results, except for station clocks at MJD 59033 and MJD 59035 epochs. As discussed in Section 2, PPP highly relies on clock products, whereas the network solution does not. Although IGSFinal and IGSRapid have the smallest daily discontinuities and are superior to the network clocks, the day-boundary discontinuities are smaller in station clock estimates obtained with network solutions. The day-boundary discontinuities in PPP station clock estimates were found to be related to the discontinuities in ACs GPS clock products. The discontinuities in these products seem to cause new errors during station clock estimates and increase the day-boundary discontinuities. By comparing Figure 3b, Figure 4b, Figure 5b, Figure 6b, Figure 7b and Figure 8b, with Figure 1, the changes in day-boundary discontinuities at different adjacent epochs seem to be correlated with the visible satellites at midnight. If the satellite clock during observation at midnight shows large discontinuities, a considerable level of day-boundary discontinuities may be observed in the PPP results. The discontinuities in precise products may be due to the discontinuities in the orbit or non-perfect overlapping strategies employed by different IGS ACs for clock generation and solution noise. Station clock estimates are likely to inherit these noises. As mentioned in Section 2.2, CODE adopts double-differenced network processing of phase observations with a 3-day-arc to estimate satellite orbits. Overlapping strategies of orbits at the boundaries superior to strategies adopted by undifferentiated GFZ. CNES-CLS employs an overlapping 6 h of data from the neighboring days to eliminate discontinuities in orbits. However, it is necessary to identify along-track orbit errors, solar radiation pressure (SRP) model errors, and other systematic or geophysical errors. Figure 3b, Figure 4b, Figure 5b, Figure 6b, Figure 7b and Figure 8b, also show that the day-boundary discontinuities are highly site-dependent due to the changes in the daily discontinuities with epochs in different station clock estimates. Although this may be caused by local code multipath errors and the temperature around the receivers, this experiment cannot indicate the problem. However, we still conclude that the day-boundary discontinuities are related to the ACs GPS products, and the daily discontinuities in ACs GPS precise products will influence the origin of the day-boundary discontinuities in PPP station clock estimates. This can be illustrated by the station clocks’ frequency stability in Figure 9a–f. Fortunately, the magnitude of the induced day-boundary discontinuity is so small that its impact can be ignored.

As shown in Figure 9a–f, the frequency stability performances of all stations clocks with IGSFinal and IGSRapid are similar, and all station clocks’ frequency stability performances with CODFinal, GFZFinal, and GRGFinal products are also similar. The frequency stability of all station clocks with IGSFinal and CODFinal products roughly performs the best with an average time of 1000 s, followed by GRGFinal, IGSRapid, and GFZFinal. We observed that the short-term frequency stability in the time series of PPP clock estimates with IGSRapid products is degraded by higher noise compared with other products. The frequency stabilities of all station clocks with IGSRapid products begin to improve slowly at an average time of 1000 s, but they degrade gradually with CODFinal and GRGFinal products, especially at an average time of more than one day. Day-boundary discontinuities seem to be ascribed to this in PPP station clock estimates with CODFinal and GRGFinal products than those with IGSRapid products during MJD 59036–59037; meanwhile, the long-term frequency stabilities of all clock estimates with IGSFinal and IGSRapid products become superior to those ACs GPS products and network solutions. This should mainly contribute to small day-boundary discontinuities in station clock estimates with IGSFinal and IGSRapid products and clock noise during later epochs of the study period. However, the long-term frequency stability performance of IPPP clock estimates with COD or GRG integer clock products should be better than those with IGSFinal at an average time of one day. Furthermore, some random-walk noises begin to appear in the station clocks of PTBB, SPT0, OP71, and USN8 with COD products from the average time of 60,000 s, leading to an upward warping trend in MDEV. This could be attributed to the new errors induced by day-boundary discontinuities.

### 4.2. The Influence of Discontinuities in Different Galileo Products on PPP Station Clock Estimates

Considering the influence of discontinuities in Galileo products on station clock estimates, several Galileo PPP experiments similar to GPS based on the MGEX products of COD and GRG were performed, and the results were compared with network solutions. Figure 10, Figure 11, Figure 12, Figure 13, Figure 14 and Figure 15 display the time series of six station clocks and the absolute values of day-boundary discontinuities within the station clocks of PTBB, WAB2, SPT0, OP71, ROAG, and USN8, respectively. A mean was removed from the time series of all station clocks, and the series was offset 0.5 ns for display purposes.

Figure 10, Figure 11, Figure 12, Figure 13, Figure 14 and Figure 15 shows that the levels of day-boundary discontinuities in all station clocks are all less than 0.3 ns for most epochs when using COD and GRG products, and there were no observed significant discontinuities for GRG and COD, although the trend in day-boundary discontinuities in the time series of each station clocks slightly changes with the employed products in PPP. We clearly observed that the amplitude of the discontinuities is highly station-dependent, and the day-boundary discontinuities for the different stations at some epochs show a large dispersion between stations, ranging from 0.01 to 0.6 ns. Higher levels of day-boundary discontinuities were observed in the station clocks of PTBB, WAB2, SPT0, and OP71 with GRG products, especially at epochs MJD 59032 and 59037. Specifically, 0.34 ns appears at MJD 59032 and 0.48 ns at MJD 59037 in the station clocks of PTBB, 0.43 and 0.47 ns in WAB2, 0.42 and 0.51 ns in OP71, 0.16 and 0.58 ns in ROAG, and 0.01 and 0.47 ns in USN8 at the same epochs. A comparison with Figure 2 shows that Galileo PPP causes a slightly higher discontinuities in GRG products in station clock estimates. Although GRG employs integer carrier-phase ambiguities fixing for clock products, such discontinuities are created due to the orbit and clock generation strategy employed by GRG for Galileo. It is also related to the stability of satellites at the adjacent epochs between daily batches. If the satellite clock is abnormal at boundary epochs, it can induce large discontinuities in station clock estimates. This is also related to the visible satellites with large discontinuities at midnight in products. Wide- and narrow-lane uncalibrated phase delay absorbed by the clock may be attributed to the origin of day-boundary discontinuities. Paying close attention to the level of day-boundary discontinuities in all station clock estimates, we found that the amplitudes of day-boundary discontinuities in the station clock estimates with COD products seem to be a litter higher than those with GRG products at MJD 59029 and 59034. In contrast, the amplitudes of day-boundary discontinuities in the station clock estimates with GRG products became higher than those with COD and GBM products at later epochs of the study period, especially at MJD 59036–59037. Additionally, Figure 10b, Figure 11b, Figure 12b, Figure 13b, Figure 14b and Figure 15b indicate that the amplitudes of day-boundary discontinuities are highly site-specific, and the discontinuity variability among stations is independent of the station clock stability but strongly related to satellite clock instability. The day-boundary discontinuities in station clocks mainly originate from code-pseudorange noise, and a small amplitude of discontinuities is ascribed to the daily discontinuities of Galileo clock products, which can be illustrated by the comparison of day-boundary discontinuities between PPP and network solutions. This indicates that Galileo clock discontinuity is contributed by a small amplitude of day-boundary discontinuities in PPP clock estimates, which is similar to GPS.

Figure 16 shows that the instabilities in the clock estimates for each station differ given the products employed and are affected by the day-boundary discontinuities. The frequency stabilities of all station clocks obtained by network solutions are superior to those obtained by PPP with Galileo COD and GRG products at an average time of one day. The inferred instability can be higher than 5 × 10^−15^ for the station clock of H-maters at the average time of one day, and all the station clocks’ instabilities can approach this performance level with COD and GRG products. Additionally, the frequency stabilities of all station clocks obtained with COD and GRG products are nearly the same. This illustrates that PPP station clock estimates are not significantly affected by products with little difference in the daily discontinuities. Random-walk noises can be observed in the long-term frequency of all clock estimates with COD and GRG products, seriously degrading station clocks’ long-term frequency stability, especially at an average time of more than one day. This agrees with our conclusion that the discontinuities in Galileo products may influence the frequency stability of PPP station clock estimates.

Since there are no combined and weighted Galileo products, their performance is different from that of GPS, and the frequency stabilities with ACs MGEX clock products are inferior to the network solutions. However, Galileo PPP clock solutions are similar to GPS PPP, indicating that the influence of day-boundary discontinuities on PPP station clock estimates through Galileo is nearly the same as through GPS. Combined and weighted MGEX products should be developed or IPPP should be used for carrier phase time transfer through Galileo.

### 4.3. The Influence of Discontinuities in Products on PPP Time Transfer

To detail whether GPS and Galileo clock discontinuities contribute to the discontinuities in time comparisons, three time links experiments were conducted, and the results were compared with network solutions. Figure 17, Figure 18, Figure 19, Figure 20, Figure 21 and Figure 22 display the results of time comparison through GPS and Galileo PPP for the three links, and the absolute values of day-boundary discontinuities within UTC(SP)−UTC(PTB), UTC(ROA)−UTC(PTB), and UTC(USNO)−UTC(PTB), respectively. The mean was removed from the time series of time comparison, and the series was offset 1ns for display purposes.

In principle, we expected the day-boundary discontinuities in PPP time comparisons through GPS and Galileo to be similar. They developed as expected, which can be found from the statistical comparison of day-boundary discontinuities in the time series of three specific time links through GPS and Galileo PPP in Figure 17, Figure 18, Figure 19, Figure 20, Figure 21 and Figure 22. In particular, the levels of day-boundary discontinuities at all epochs of MJD 59029 to 59037 in the three time links are nearly the differences in the discontinuities in station clock estimates. If the day-boundary discontinuities in the clock estimates of two stations are similar for the same epochs, then the day-boundary discontinuities’ level will decrease in the time comparison results for these epochs. Otherwise, the level of discontinuities will be increase. That is, if the sign of the day-boundary discontinuities in two station clock estimates are the same, the discontinuities will be reduced, and vice versa. Using the time comparison of UTC(USNO)−UTC(PTB) through Galileo PPP as an example, the day-boundary discontinuities at MJD 59029 in USN8 clock estimates with COD and GRG products are −150 and 281 ps, respectively, and the day-boundary discontinuities in the PTBB clock estimates with the above products at the same epochs are −207 and 257 ps, respectively. As a result, we concluded that the magnitude of day-boundary discontinuities in the time comparison of UTC(USNO)−UTC(PTB) at the same epochs are about 56 and 24 ps with COD and GRG products, respectively. Similarly, for GPS PPP with CODFinal, GFZFinal, and GRGFinal products at MJD 59029, the day-boundary discontinuities in USN8 clock estimates are −82, 413, and 420 ps, respectively, whereas the magnitude of day-boundary discontinuities in PTBB clock estimates with the above products are 66, 286, and 348 ps, respectively, and the magnitudes of the day-boundary discontinuities in the time comparison of UTC(USNO)−UTC(PTB) for the same epoch are about 148, 117, and 71 ps, respectively. The same phenomenon can be seen at other epochs. Similar situations also exist for the time comparisons of UTC(SP)−UTC(PTB) and UTC(ROA)−UTC(PTB). The day-boundary discontinuities in the time comparisons through Galileo PPP are comparable with those through GPS. Hence, if day-boundary discontinuities in station clock estimates induced by the daily discontinuities of products are inherited by time comparisons, the discontinuities in time comparisons will deteriorate. However, the influence of the daily discontinuities of current Galileo clocks on PPP time comparisons is similar to GPS, which is not particularly critical to time comparison.

By comparing the frequency stability for the three time links displayed in Figure 23a–c and Figure 24a–c, we concluded that the short-term frequency of GPS PPP with IGSFinal products is the most stable, followed by those with CODFinal, GFZFinal, GRGFinal, and IGSRapid products. However, the frequency stability with IGSFinal and IGSRapid isnearly superior to those with CODFinal, GFZFinal, and GRGFinal products at an average time of more than one day. These results agree with the results obtained in Section 4.1. The frequency stability of Galileo PPP with COD products is superior to those with GRG products, especially at an average time of more than one day. The short- and long-term frequency stabilities of all time links degrade gradually with the GRG products due to the errors induced by the high level of day-boundary discontinuities. The day-boundary discontinuities present random-walk noise characteristics and seriously affect station clocks’ long-term frequency stability, especially at an average time of more than one day. We concluded from the above analysis that once station clock estimates inherit the discontinuities in precise products, the day-boundary discontinuities in the time comparison results increase, which will affect the short- and long-term frequency stability of PPP time comparison.

## 5. Summary and Conclusions

In this paper, the influence of discontinuities in precise products on PPP clock estimates and time comparisons was investigated to understand the origin of day-boundary discontinuities in PPP station clock estimates and time comparisons. Several experiments of station clock estimates through GPS PPP and Galileo PPP with products provided by several IGS and MGEX ACs were performed. The statistics of day-boundary discontinuities were then compared to the network processing solution results. The frequency stability of station clock estimates and time comparisons were also used in our analysis.

Some conclusions were obtained from this study. First, considering the processing strategies of IGS combined products, the daily discontinuities in IGSFinal and IGSRapid products are smaller than 0.1 ns, ascribed to clock instability and negligible to the day-boundary discontinuities in the time comparison. However, the daily discontinuities in ACs GPS clocks are larger, and these discontinuities seem to be inherited by PPP station clock estimates and, more importantly, contributing to the origin of the day-boundary discontinuities in GPS PPP station clock estimates. The short- and long-term frequency stabilities of PPP station clocks also slightly change with the ACs GPS products employed, and degrade due to the daily discontinuities in GPS products. Secondly, although the daily discontinuities in Galileo clocks are larger than those in GPS clocks, their influence on the day-boundary discontinuities in Galileo PPP station clock estimates is nearly the same as in GPS PPP. The daily discontinuities in Galileo clock products contribute a small amplitude to the day-boundary discontinuities in PPP station clock estimates. The generating strategies adopted by different ACs for orbit and clock products and clocks’ problems together should be attributed to the Galileo clock daily discontinuities. The day-boundary discontinuities present random-walk noise characteristics, which seriously affects station clocks’ long-term frequency stability, especially at an average time of more than one day. Finally, the influence of daily discontinuities in Galileo clocks on PPP time comparison is similar to GPS and is not particularly critical to time comparison. However, combined and weighted MGEX products should be developed, or Galileo IPPP should be used for remote comparison of high stability clocks.

The other factors contributing to the day-boundary discontinuities in GNSS PPP time transfer should be investigated more carefully, especially for time comparison through new GNSS constellations like BDS. ISBs will be carefully characterized by different GNSS clock products. Developing effective methods for eliminating day-boundary discontinuities in time comparisons of PPP through new GNSS constellations should be considered a future research topic.

## Figures and Tables

**Figure 1 sensors-21-01156-f001:**
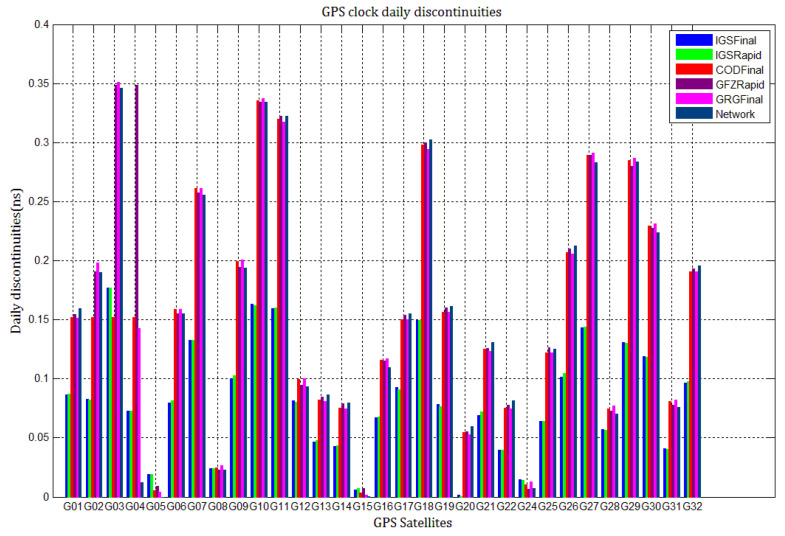
The daily discontinuities in IGSFinal, IGSRapid, CODFinal, GFZFinal, GRGFinal, and network global positioning system (GPS) clocks during MJD59029 to 59037.

**Figure 2 sensors-21-01156-f002:**
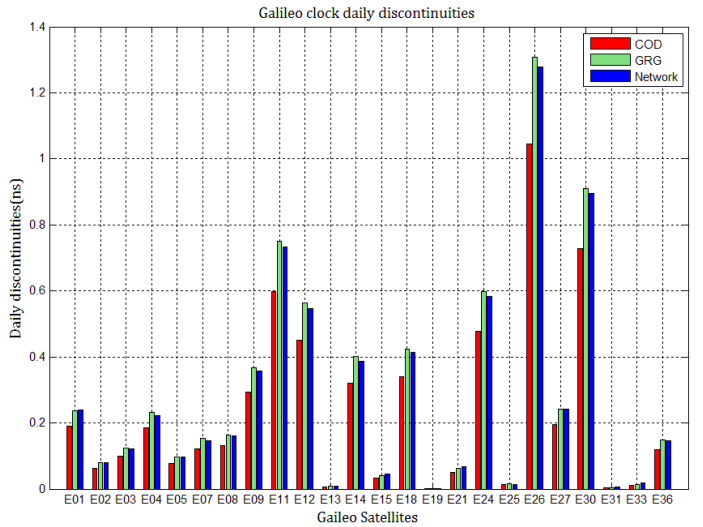
The average absolute values of daily discontinuities in COD, GRG, and network Galileo clocks during MJD 59029 to 59037.

**Figure 3 sensors-21-01156-f003:**
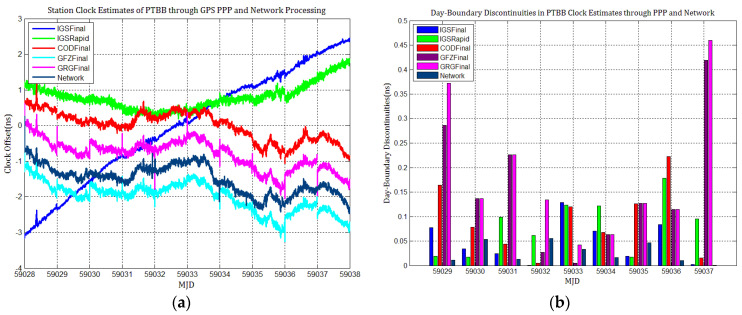
(**a**) PTBB station clock estimates through GPS PPP and network solutions and (**b**) the statistical results of the day-boundary discontinuities.

**Figure 4 sensors-21-01156-f004:**
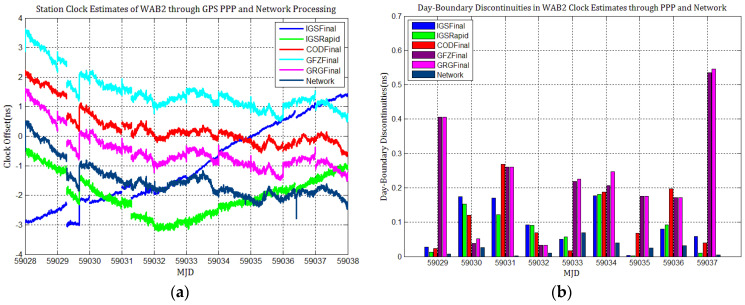
(**a**) WAB2 station clock estimates through GPS PPP and network solutions; (**b**) day-boundary discontinuities in clock estimates.

**Figure 5 sensors-21-01156-f005:**
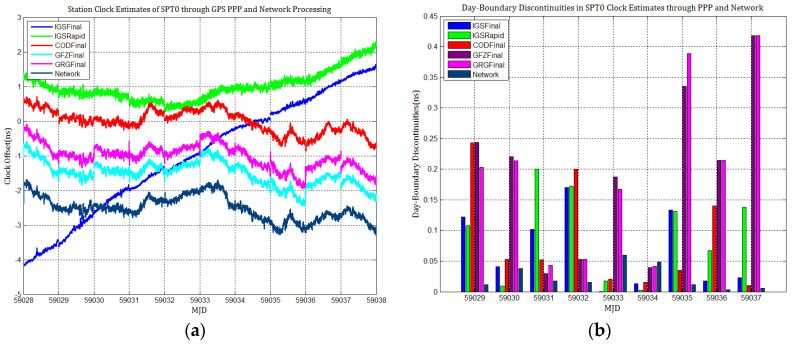
(**a**) SPT0 station clock estimates through GPS PPP and network solutions; (**b**) day-boundary discontinuities in clock estimates.

**Figure 6 sensors-21-01156-f006:**
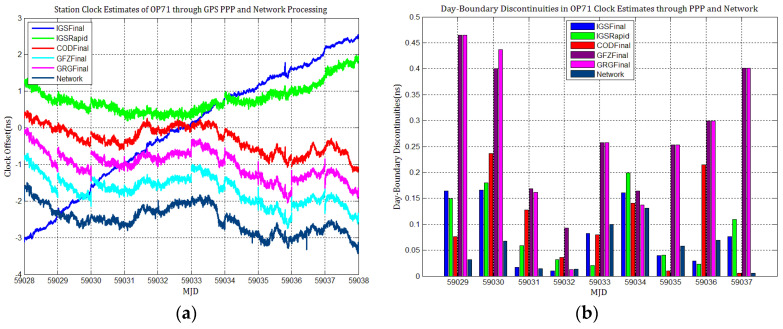
(**a**) Station clock estimates of OP71 through GPS PPP and network solutions; (**b**) day-boundary discontinuities in clock estimates.

**Figure 7 sensors-21-01156-f007:**
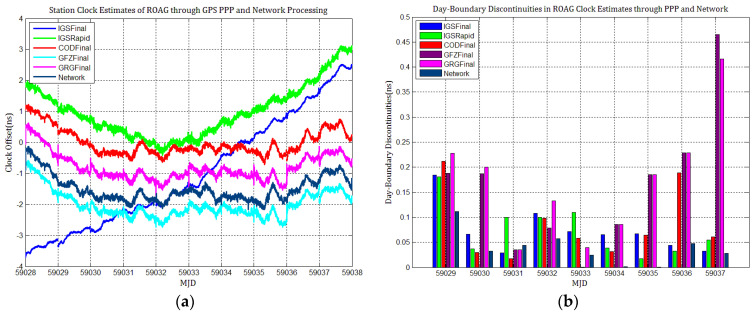
(**a**) ROAG station clock estimates through GPS PPP and network solution; (**b**) day-boundary discontinuities in clock estimates.

**Figure 8 sensors-21-01156-f008:**
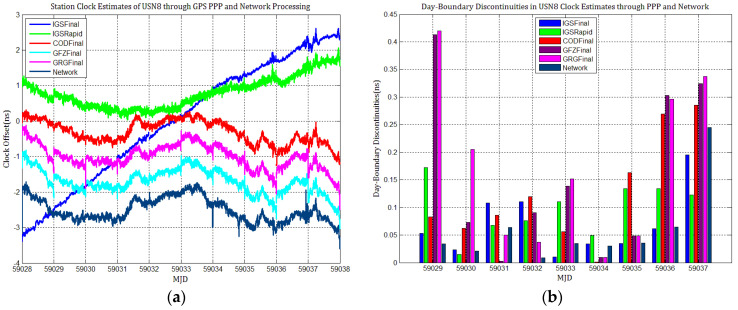
(**a**) USN8 station clock estimates through GPS PPP and network solution; (**b**) day-boundary discontinuities in clock estimates.

**Figure 9 sensors-21-01156-f009:**
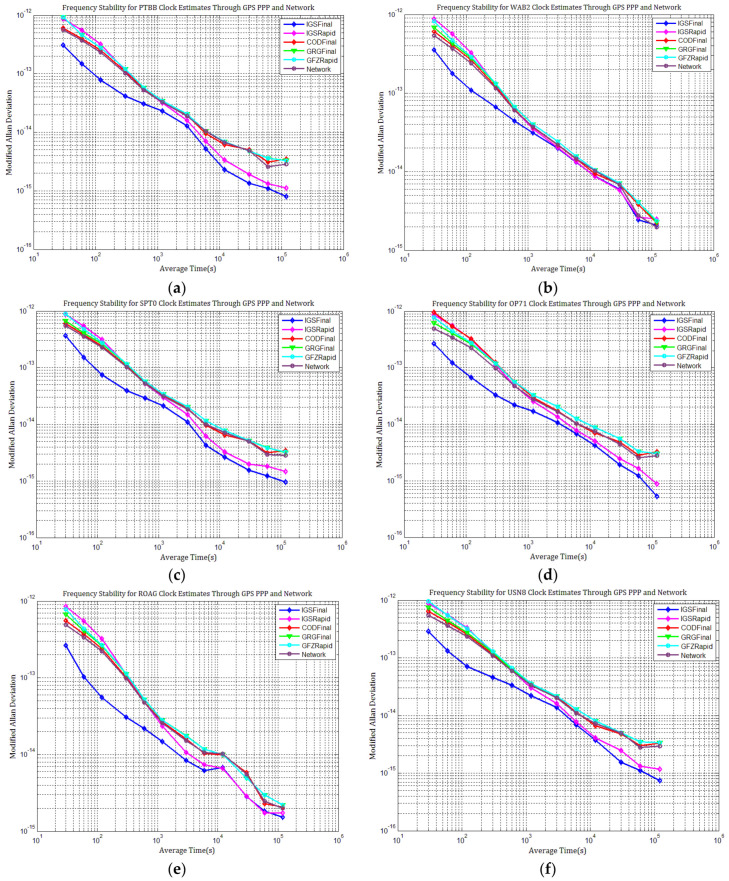
Frequency stability of station clock estimates for PTBB, WTZS, SPT0, WAB2, OP71, ROAG, and USN8 through GPS PPP with different products: (**a**) MDEV of PTBB clock estimates; (**b**) MDEV of WAB2 clock estimates; (**c**) MDEV of SPT0 clock estimates; (**d**) MDEV of OP71 clock estimates; (**e**) MDEV of ROAG clock estimates; (**f**) MDEV of USN8 clock estimates.

**Figure 10 sensors-21-01156-f010:**
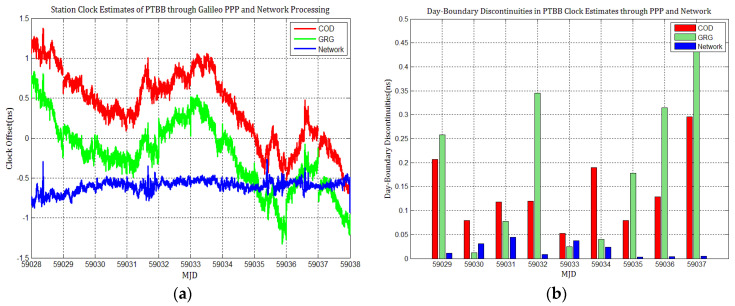
(**a**) PTBB station clock estimates through Galileo PPP and network solutions; (**b**) day-boundary discontinuities in station clock estimates.

**Figure 11 sensors-21-01156-f011:**
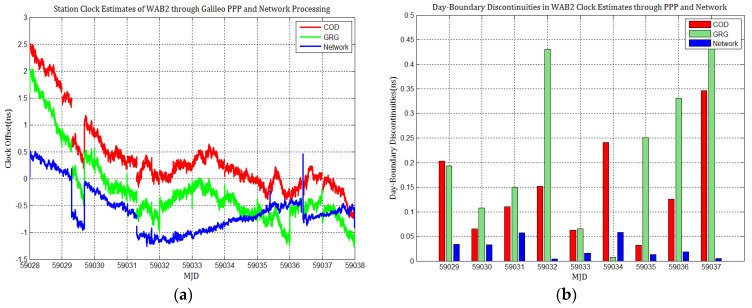
(**a**) WAB2 station clock estimates through Galileo PPP and network solution; (**b**) day-boundary discontinuities in station clock estimates.

**Figure 12 sensors-21-01156-f012:**
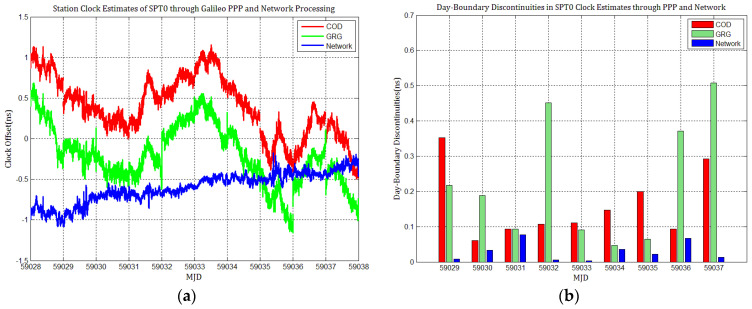
(**a**) SPT0 station clock estimates through Galileo PPP and network solution; (**b**) day-boundary discontinuities in station clock estimates.

**Figure 13 sensors-21-01156-f013:**
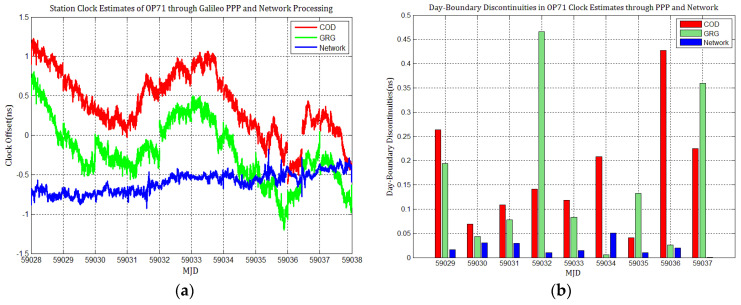
(**a**) OP71 station clock estimates of through Galileo PPP and network solution; (**b**) day-boundary discontinuities in station clock estimates.

**Figure 14 sensors-21-01156-f014:**
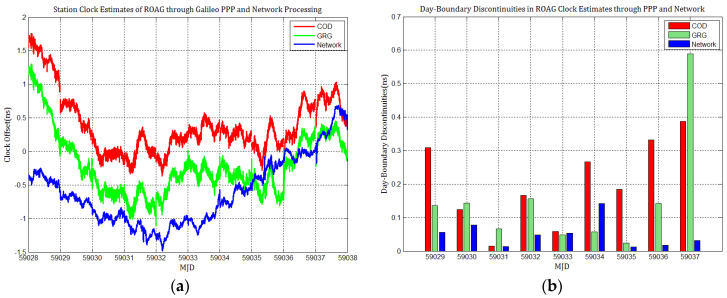
(**a**) ROAG station clock estimates through Galileo PPP and network solution; (**b**) day-boundary discontinuities in station clock estimates.

**Figure 15 sensors-21-01156-f015:**
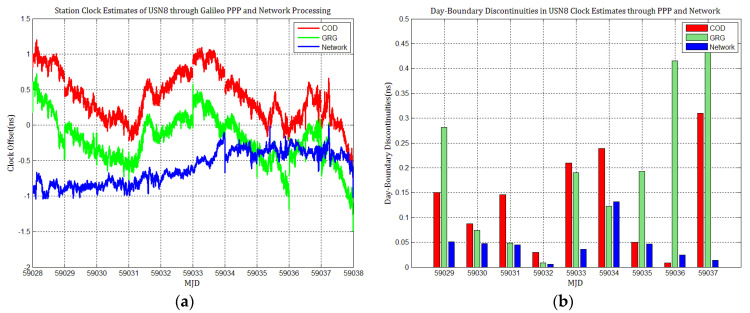
(**a**) USN8 station clock estimates through Galileo PPP and network solution; (**b**) day-boundary discontinuities in station clock estimates.

**Figure 16 sensors-21-01156-f016:**
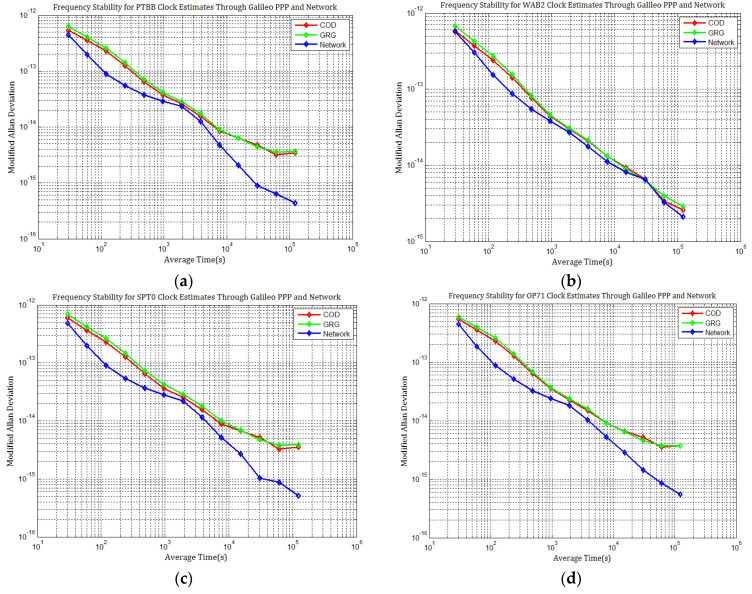
Frequency stability of station clocks through Galileo PPP with different products: (**a**) MDEV of PTBB clock estimates; (**b**) MDEV of WAB2 clock estimates; (**c**) MDEV of SPT0 clock estimates; (**d**) MDEV of OP71 clock estimates; (**e**) MDEV of ROAG clock estimates; (**f**) MDEV of USN8 clock estimates.

**Figure 17 sensors-21-01156-f017:**
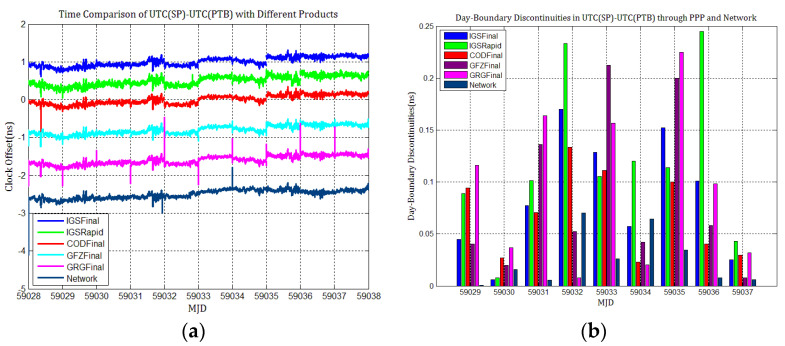
(**a**) Time comparison of UTC(SP)−UTC(PTB) through GPS PPP; (**b**) day-boundary discontinuities in time comparisons.

**Figure 18 sensors-21-01156-f018:**
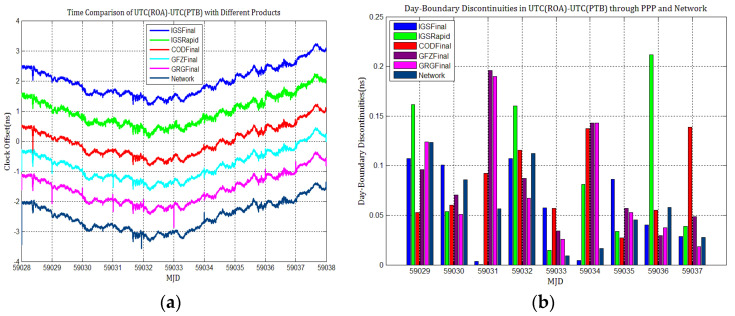
(**a**) Time comparison of UTC(ROA)−UTC(PTB) through GPS PPP; (**b**) day-boundary discontinuities in time comparisons.

**Figure 19 sensors-21-01156-f019:**
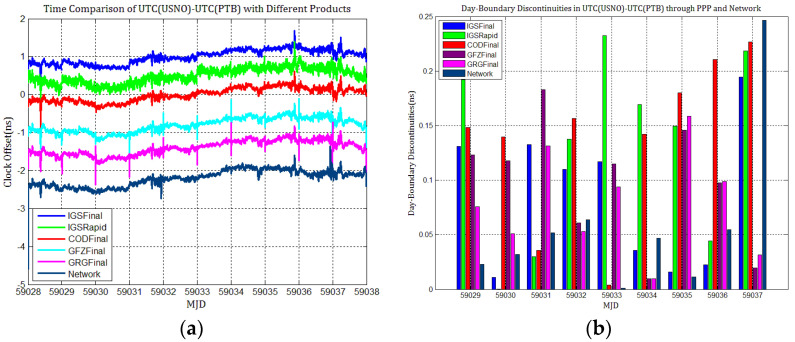
(**a**) Time comparison of UTC(USNO)–UTC(PTB) through GPS PPP; (**b**) day-boundary discontinuities in time comparisons.

**Figure 20 sensors-21-01156-f020:**
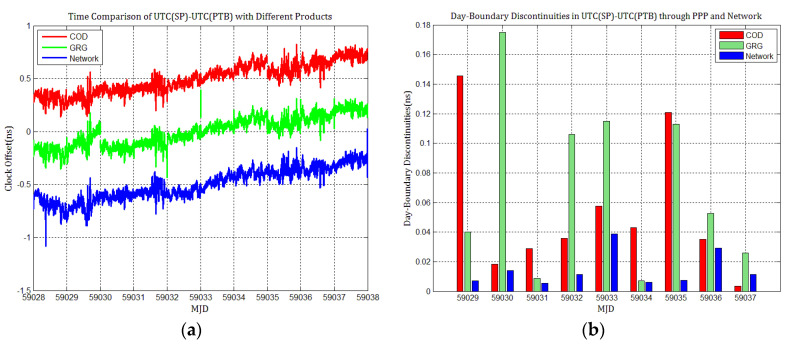
(**a**) Time comparison of UTC(SP)–UTC(PTB) through Galileo PPP; (**b**) day-boundary discontinuities in time comparisons.

**Figure 21 sensors-21-01156-f021:**
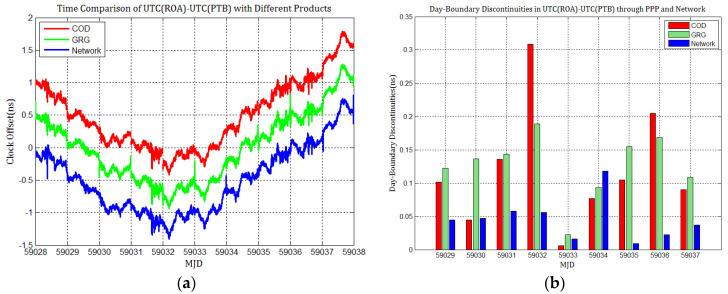
(**a**) Time comparison of UTC(ROA)–UTC(PTB) through Galileo PPP; (**b**) day-boundary discontinuities in time comparison.

**Figure 22 sensors-21-01156-f022:**
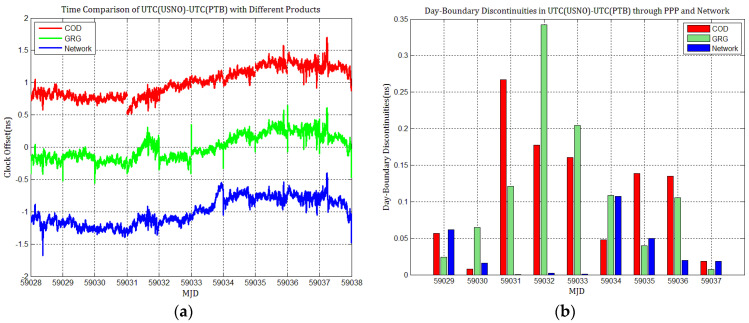
(**a**) Time comparison of UTC(USNO)– UTC(PTB) through Galileo PPP; (**b**) day-boundary discontinuities in time comparison.

**Figure 23 sensors-21-01156-f023:**
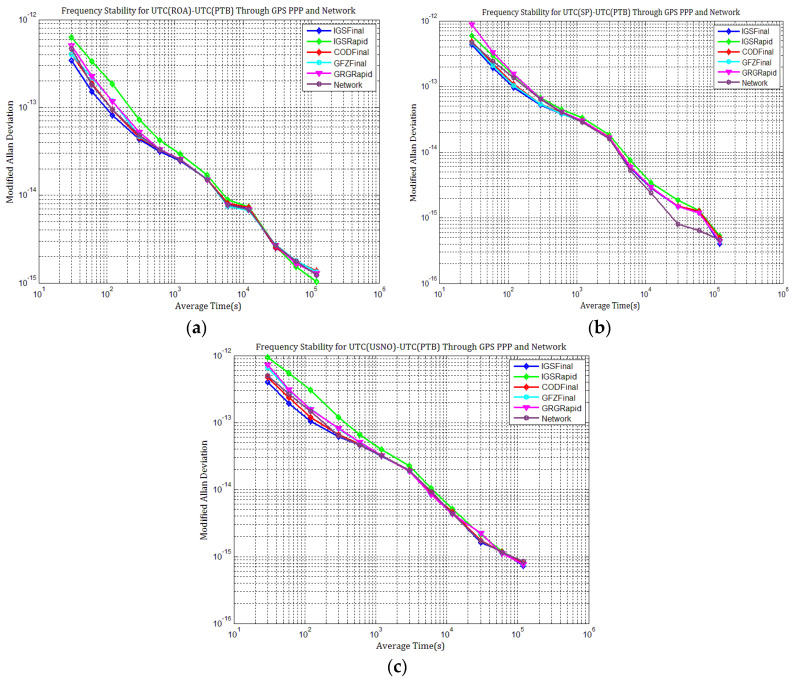
Frequency stability of UTC(SP)–UTC(PTB), UTC(ROA)–UTC(PTB), and UTC(USNO)–UTC(PTB) through GPS PPP with different products: (**a**) MDEV of UTC(SP)–UTC(PTB); (**b**) MDEV of UTC(ROA)–UTC(PTB); (**c**) MDEV of UTC(USNO)–UTC(PTB).

**Figure 24 sensors-21-01156-f024:**
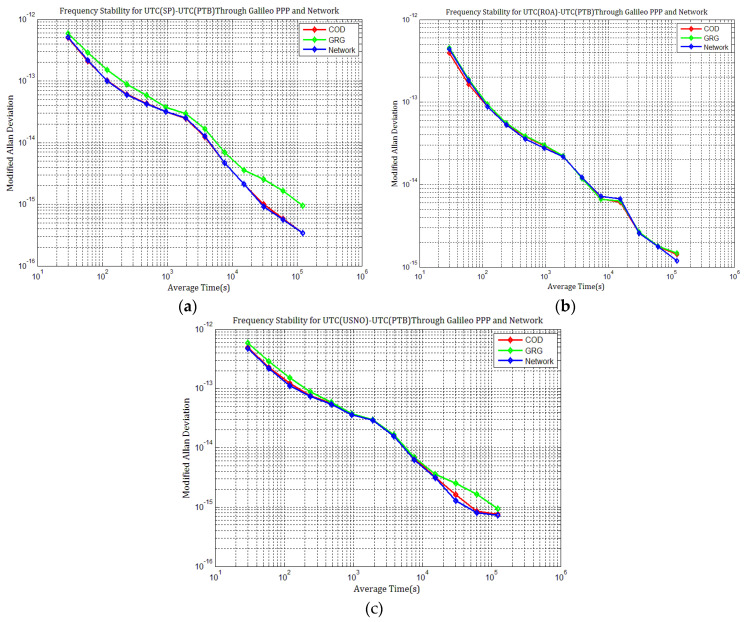
Frequency stability of UTC(USNO)–UTC(PTB), UTC(ROA)–UTC(PTB), and UTC(USNO)–UTC(PTB) through Galileo PPP with different products: (**a**) MDEV of UTC(SP)–UTC(PTB); (**b**) MDEV of UTC(ROA)–UTC(PTB); (**c**) MDEV of UTC(USNO)–UTC(PTB).

**Table 1 sensors-21-01156-t001:** Ground stations and equipment configurations.

Station	Receiver	Country	Antenna	External Reference
PTBB	SEPT POLARX5TR	Germany	LEIAR25.R4	UTC(PTB)
SPT0	SEPT POLARX5TR	Sweden	LEIAR25.R4	UTC(SP)
WAB2	SEPT POLARX5TR	Switzerland	SEPCHOKE_B3E6	UTC(CH)
OP71	SEPT POLARX4TR	France	LEIAR25.R4	UTC(OP)
ROAG	SEPT POLARX5TR	Spain	LEIAR25.R4	UTC(ROA)
USN8	SEPT POLARX5TR	USA	TPSCR.G5	UTC(USNO)

## Data Availability

All data can be available from the international GNSS Monitoring and Assessment System (iGMAS), http://www.igmas.org.

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
