# Peer review of "Influence of Precise Products on the Day-Boundary Discontinuities in GNSS Carrier Phase Time Transfer"

_sensors, 2021, doi:10.3390/s21041156_

Round 1

Reviewer 1 Report

The authors of the manuscript “Influence of precise products on the day-boundary discontinuities in GNSS carrier phase time transfer” studied the effects of the products provided by IGS and by IGS Analysis Centers (ACs) on the estimate of GNSS carrier phase time transfer using the Precise Point Positioning (PPP) approach, focusing on the change of day or block o days. This study was applied to both GPS and Galileo system. To quantified this effect a network solution is computed and compared with the PPP solutions, obtained using IGS final products and the ones provided by IGS ACs.

In my opinion the conclusions of the authors are supported by the results. The only observation I feel to do is to show the correlation of the discontinuities with the visible satellites at midnight, as mentioned at lines 326-328.

Minor remarks:

  1. check the space before the citation in the manuscript. For example: at line 54 “code-pseudorange[12–15]” —> “code-pseudorange [12–15]”, and so on.
  2. Line 44: Time Coordinated(UTC), space between Coordinated and (UTC).
  3. CiteTable 1 and 2 into the text

Author Response

Thank you very much for the constructive and valuable comments of the reviewer. We have carefully revised all the problems involved in the manuscript, which are as follows: 1. As for "check the space before the citation in the manuscript. For example: at line 54 “code-pseudorange[12–15]” —> “code-pseudorange [12–15]”, and so on." Thanks for the carefulness of the reviewer. Yes, as comments of the reviewer, the full text in the manuscript has been carefully checked and all mistakes like this have been revised, including at line 16 "frequency transfer [1-2]", at line 53 "code-pseudorange [12-15]", at line 58" phase-only [18-22]", at line 74 "individual ACs [23-25]", at line 84 " day-boundary discontinuities [26]", at line 137 "IGS combinations [28, 29]", at line 152 " be noticed [30]", at line 171 " orbit solution [32, 33]", at line 194 " clock products [39]", at line 226 " at midnight [32]" and at line 237 " was proposed [23]". 2. As for " Line 44: Time Coordinated(UTC), space between Coordinated and (UTC)." Yes, we firmly believe that it is another kind of mistakes in full text of manuscript, and all such mistakes have been revised as reviewer's comment, including at line 16 "Satellite System (GNSS)", at line 44 " Time Coordinated (UTC)", at line 407 " clock estimates; (d)", and at line 513 "clock estimates; (d)". 3. As for " CiteTable 1 and 2 into the text." Yes, of course, according to the reviewer’s comments and referring to the rules of the paper template, all tables in the manuscript have been revised and cited into the text. 4. As for the point that the reviewer care about "the correlation of the discontinuities with the visible satellites at midnight, as mentioned at lines 326-328. " From Figure 1 to Figure 3(b)-8(b) in the manuscript, we can clearly see that the day-boundary discontinuities varies as the stations and amplitudes of day-boundary discontinuities in different adjacent epochs may be correlated with the visible satellites at midnight, this can be possible illustrated from the changes of day-boundary discontinuities in stations in Europe (PTBB, WAB2, SPT0, OP71 and ROAG) and the station in North-America. Moreover, it can been illustrated from another perspective, that is, if we remove the satellites with a large discontinuity in orbit and clock products and the day-boundary discontinuities can be reduced. We will further describe this problem from other perspectives if possible.

Reviewer 2 Report

Precise Point Positioning (PPP) is a technique used to determine highaccuracy position with a single GNSS receiver. May be an alternative solution to differential measurements, where main-taining a connection with a single Real Time Kinematic (RTK) station or a regional Real Time Network (RTN) of reference stations is necessary. This situation is especially common in areas with poorly developed infrastructure of ground stations.
In the paper "Influence of precise products on the day-boundary discontinuities in GNSS carrier phase time transfer", the Authors investigated the influence of GNSS precise products on the day-boundary discontinuities in PPP station clock estimates and time comparison, with the aim of better understanding the source of the day-boundary discontinuities.
The structure of the article is considered and clear. In the introduction, background and review of the problem's literature were presented.
The main part of the article presents principle of PPP, GPS and Galileo precise products and the problem of discontinuities in clock products and network processing solution to eliminate day-boundary discontinuities in PPP. Conclusions, on the basis of the research, are clear.
The article is very interesting and has high scientific advantages.
It needs to be editoral corrected:
Line 16: ... System (GNSS) - with space - this editoral mistake should be corrected in many places
Line 21: Abbreviation can be used: 0.3ns
Line 27: The abbreviation MJD should beb explained
Line 45: Abbreviation can be used: 100ps
Line 130: (ISB) rather than ( ISB ) - with no spaces
Line 153: noticed [30] with space rather than noticed[30]
Line 203: 76.8ps or 76.8ns, probably 76.8ps rather than 76.8pns
Line 486: 5⋅10 rather than 5x10
Line 489: as above
Line 529: -0.06ns - is tie negative?
Line 531, 533, 535, 538: as above
Line 636: GRG, with no space

Author Response

Thank you very much for the constructive and valuable comments of the reviewer. We have carefully revised all the problems as mentioned in the manuscript, which are as follows:

  1. As for " ... System (GNSS) - with space in line 16- this editoral mistake should be corrected in many places".

Thanks for the carefulness of the reviewer. Yes, there are many such mistakes in the full text of manuscript and we have checked carefully all mistakes and then revised as the reviewer's comments, including at line 16 " Global Navigation Satellite System (GNSS)", at line 44 " Universal Time Coordinated (UTC)", at line 407 " clock estimates; (d)", and at line 513 "clock estimates; (d)" and the same problems in all tables.

  1. As for "Abbreviation can be used: 0.3ns in line 21".

Yes, as recognized by all researchers of time and frequency transfer, "0.3 nanoseconds" can be abbreviated as "0.3ns", we would like to use the abbreviation of "ns" and "ps" in the full text of manuscript instead of " nanoseconds" and "picoseconds" if there is no problem for editorial and publication. All expressions of nanoseconds and picoseconds have been abbreviated as the comments, including at line 21 "0.3 nanoseconds" and line 45 "100 picoseconds".

  1. As for " Line 27: The abbreviation MJD should be explained".

Yes, we checked carefully and explain MJD, which is the abbreviation of Modified Julian Date. Moreover, we have made a detailed check to the manuscript to avoid such mistakes.

  1. As for " Line 130: (ISB) rather than ( ISB ) - with no spaces".

Yes, we firmly believe that it is another kind of mistakes in full text of manuscript, and all such mistakes have been revised as reviewer's comment, including at line 130 " inter-system bias (ISB)", at line 614 "UTC(OP)" and line 618 " UTC(USNO)". Furthermore, a detailed inspection to the manuscript has been performed in order not to occur such mistakes.

  1. As for "Line 203: 76.8ps or 76.8ns, probably 76.8ps rather than 76.8pns".

Please forgive the mistake and we are sure that it is 76.8ps. This should be a mistake in the revision of the paper by our different authors. Thanks for the carefulness.

  1. As for " Line 486: 5∙10 rather than 5x10".

We checked the manuscript and made sure there are several expressions in the part of "Result and Analysis", like " 5x10−15" at line 485 and " 4x10−14" at line 488. Such expressions have been revised as the reviewer's comments.

  1. As for "-0.06ns - is tie negative? ".

We checked such problems from the line 528 to 537 and make sure there are due to incorrect notations during using the "Microsoft Word". Then we have revised all incorrect notations as the reviewer's comments.

  1. As for "Line 636: GRG, with no space."

We have checked carefully in the manuscript and revised it. It is really difficult to find it and many thanks for reviewer's carefulness.

Reviewer 3 Report

The paper is written in a poor, almost broken English and at times it is even difficult to understand what was meant. The English should be first checked and corrected before sending out for reviews. Consequently, only a cursory evaluation/review is provided below:

In my opinion, the paper provides a little or no new information, is rather long and yet it does not even give some relevant and important information, e.g., on the PPP software used and network processing, advocated and presented here; it also contains some a wrong and/or incomplete information (e.g., on IGS/MGEX solution products). Furthermore, it is written in a rather poor English.

The possible daily discontinuities in IGS Final products are small and mostly negligible in most cases (contrary to the claims made here) and are significantly attenuated (averaged) in PPP's. They can be nearly eliminated either by multi-day PPP's or by a daily PPP with Ambiguity Resolution. (AR PPP). In particular it is so for ARPPP's with continuous AR integer cycle ambiguity datum maintained between days, as it is currently, done by BIPMS for UTC determination. This not only removes the daily discontinuities, but also increases the time transfer precision as well as the long term frequency stability of AR PPP clock solutions by an order of magnitude with respect to float PPP's. Note that COD, GRG (and likely GFZ/GBM) already routinely provide the bias information enabling AR PPP and so does IGS on an experimental bases only (see the IGS PPP Working Group) . With AR PPP there is little or no advantage of a network processing, besides, no network clock solutions is presented here, only the network daily discontinuities, why?

For these reasons the paper should be rejected, in my opinion and if a resubmission is encouraged, the first and foremost the English should be improved, the paper shortened, PPP and network software/solutions properly described, network clock solution presented along with the AR PPP's also included

Below there are a few additional comments/suggestions:

Eqs 1 & 2: the j index needs to be described, too

Tabs 1 & 2: should be carefully checked and corrected

Figs 1 & 2: What is this? Standard deviations or RMS? This should be spelled out in the legend.
Also note that since the IGS/MGEX orbits/clock daily products, by the IGS convention, do not contain overlapping epochs, (i.e. there is a 30 sec interval between the last epoch of the previous day and the first epoch of the next day), the most GPS 'discontinuities' are due to GPS clock instability over the 30 sec interval as well as the formal solution errors of the two epochs. This is confirmed by COD clocks, which are continuous (over 3 days). The clock instability problem is also apparent for the Gal E11 Rb clock in Fig. 2

2.3. Network Processing Solution: needs to be described. What orbits are used? IGS or AC Final/MGEX ones? According to the text no IGS/MGEX products (orbits/clocks) are used. This is very doubtful, or impossible to solve for the satellite orbits/clocks from only a few (6 non globally distributed stations), and even with better solution accuracy and daily orbit/clock discontinuities than IGS or ACs? This needs to be described and full clock solutions also presented in all Figs, not just the network discontinuities.

Are Galileo-only PPP used? Some MGEX ACs (like CODE) , unlike for GPS, use a different, non standard antenna PCO/PCV atx file, this non standard Galileo PCO/PCV must be used for PPP clock solutions

Fig. 16 the plot for GBM frequency stability at 10E5 sec must be wrong, it should be worse due to the large daily discontinuities ...

References: Some citations of references is not correct/appropriate, i.e. the references do not correspond to the text citations, please check

Author Response

Thank you very much for the constructive and valuable comments of the reviewer. We have carefully revised all the points involved in the manuscript as the reviewer's suggestions, which are as follows:

  1. As for " the first and foremost the English must be corrected and improved. "

We are very grateful to the review for the professional comments and good suggestions about the English language. Yes, the improvement of English language is the significant problem we need to resolve. As the reviewer's requirements, the full text in the manuscript have been checked and corrected, including sentence structure, grammar, acronym definition and English spelling, especially the reversions in the parts of 'Abstract', 'Introduction', 'Result and Analysis', 'Summary and Conclusion', which could be viewed from the annotations of the modified version of my manuscript. Specifically, in 'Abstract', we checked all of the acronym definitions and added the definition of 'Modified Julian Date (MJD)'. We reorganized the sentences, and simply introduce background of day-boundary discontinuities in PPP time transfer. Additionally, we emphasize the residual day-boundary discontinuities in PPP station clock estimates and time comparisons through new GNSSs like Galileo seems larger than those in GPS PPP after elimination, especially using precise clock products with large discontinuities. Of course, we also introduced the aim of network processing solution results.  

In part of 'Introduction' from line 16 to 43, firstly, the sentences have been reformulated thoroughly in order to clearly introduce the aiming we presented in the paper. Secondly, a lot of contents are added to enrich the background, including the IGS/BIPM pilot project to exploit geodetic time and frequency transfer, the traditional methods of Common-view and TWSTFT and their disadvantages, GPS IPPP technique and its advantages in eliminate the day-boundary discontinuities & applications in time and frequency transfer for TAI/UTC computation, current research status of PPP time comparison through new GNSSs like Galileo and Beidou, shortage of IPPP through new GNSSs, and background of MGEX products. Thirdly, the order of the sentences has also been reformulated in order to clearly state what we would like to express.

In part of 'Principle and Methods' , as the reviewer's comments, firstly we presents the details of relevant PPP principle in our data procession, which could be clearly viewed from line 128 to 162 of this paper. Secondly, with paying close attention to the updated generating strategies that IGS and MGEX ACs adopted and published online through papers, we carefully checked and corrected the contents of GPS and Galileo precise products and the problem of discontinuities in clock products provided by all related ACs, especially the incorrect contents in the table 1 and table 2. We canceled out these tables after a careful consideration, and presents relating contents in the text, which can be clearly viewed from line 177 to 210 of this paper. Meanwhile, we clearly illustrate the average absolute value of day-boundary discontinuities that Figure 1 and Figure 2 pointed to, which can be viewed from line 220 to 244 of this paper, and Figures of 1 and 2. Thirdly, we described the details of network processing solution we used in this paper and pointed out why we choose it rather than IPPP to illustrate the effects of the precise products on day-boundary discontinuities. Moreover, we adjusted the orders of related sentences to present the advantages of network solution in eliminating day-boundary discontinuities, which can be viewed from line 249 to 273 of this paper. Finally, we shorted but more clearly presented the performs of discontinuities in 5 kinds of GPS precise clock products and 3 kinds of MGEX products, of course including the induced reasons due to generating strategies ACs currently adopted.

In part of 'Experiment Design and Data Processing', we corrected the description of network processing solution, and added the introduction of the set of stations worldwide we are interested and the IGS/MGEX orbits we used in order to explain the reviewer’s doubts, which can be viewed from line 275 to 293. Meanwhile, we revised the network experiments design and illustrated our consideration about the impact of ISBs to our experimental results, as the line of 305-308 shows.

In part of 'Results and Analysis', Firstly, we canceled out the unnecessary descriptions of our experimental results in the front section of this part. The most of important, as the reviewer's comments, we added the results of station clock estimates of network solution in all figures (3-16), not just the network discontinuities. However, we kept the figures of 17-22 as original in the manuscript because our concerns are the difference of time transfer through Galileo PPP with different MGEX products, therefore, there is no use of network solution results for comparison. Secondly, we reorganized the relevant sentences about the specific experiment results through PPP station clock estimates with different GPS products and network solution in section of 4.2, especially the statistical results of day-boundary discontinuities in station clock estimates with IGSFinal, IGSRapid, CODFinal, GRGFinal and GFZFinal products at stations of PTBB, WAB2, SPT0, OP71, ROAB and USN8, at epochs of MJD 59029 to 59037., which can clearly viewed from line 311 to 356. We have corrected the sentences too subjective to be consist with experiment results in this section. Additionally, the analysis of frequency stability of GPS PPP station clock offsets are also corrected. Thirdly, we modified the description of Galileo PPP station clock estimates with products of COD, GBM and GRG and the analysis of frequency stability, and revised our original subjective conclusions according the actual experiment results, which are shown in lines of 392 to 409. Furthermore, we have made a lot of appropriate modifications to the sentences of experiments results and analysis of influence of discontinuities in MGEX clock products on Galileo PPP time comparison, frequency stability of clocks and double-differenced results of typical three time links, to enhance the logic of sentences.

In the last part ' Summary and Conclusions', we checked the sentences of conclusions thoroughly and a lot of appropriate modifications to the sentences of conclusions have been made in order to properly make the conclusions more supported by the experiment results. Also, the conclusion after revision is obviously more consistent with abstract.

  1. As for " PPP and network software/solutions properly described, network clock solution presented along with the AR PPP's also included. "

Yes, we have revised and gave the rigorously description of relevant PPP and network processing solution we developed in our data procession in section 2.1 and 2.3. AR PPP or IPPP, proposed by Centre National d'Etudes Spatiales (CNES) and Collecte Localisation Satellites (CLS), can be used for recovering the integer ambiguity and cancelling out the day-boundary discontinuities. It can achieve continuous time transfer through IPPP, and the stability of frequency comparisons at an averaging time of a few hours and above, especially more than 5×10−16 at 1d averaging time can be improved. Network solution with stations worldwide, can also achieve an continuous time comparisons and high frequency stability. However, as we stated in introduction and section 2.3 of this paper, although both network clock solution and IPPP have advantages and are effective techniques to improve the stability of frequency comparisons, IPPP are highly rely on wide-lane satellite phase bias products, and it is currently only adopted GPS for UTC because it is not all MGEX ACs in this paper could provide such phase bias products for new GNSSs, especially Galileo and BeiDou. For example, as Prange, L et.al(2020), the COD analysis performs NL and WL AR for GPS and Galileo, it demonstrates that phase-aligned COD Galileo clocks in combination with OPBs allow for PPP with ambiguity fixing (PPP-AR). However, ambiguity-fixing does not improve the day-to-day continuity of satellite clocks estimated in independent daily sessions, and even increase their value for dedicated applications. Meanwhile, it is difficult for us to apply ambiguity fixing to zero-difference network solution in a short time. Moreover, our main focus are investigating the effect of precise GPS or Galileo products on the origin of day-boundary discontinuities in classical PPP station clock estimates and time comparisons, and it is better to compare classical PPP depended on GNSS precise products with a method without highly rely on GNSS precise products, so it is enough and suitable to use network solution results for comparison and demonstrate our conclusions. Therefore, we only presented the experiments of network solution with float ambiguities rather than both of network solution and AR PPP or AR network solution. Our further research will be focus on network processing with integer carrier-phase ambiguity resolution using combined multi-GNSS observations, especially GPS/ Galileo/BDS.

  1. As for " Figs 1 & 2: What is this? Standard deviations or RMS? This should be spelled out in the legend."

Yes, we carefully checked related sentences and acknowledge that the specific contents in figure 1 and 2 are not clearly stated. It is the average absolute value of discontinuities in 5 kinds of GPS-specific clock products and 3 kinds of Galileo-specific clock products during the period of 10 days (MJD 59029 to 59037) , and we have revised the descriptions of these two figures. These discontinuities in clock products are calculated as the average absolute value of the last 5 minutes of the last day's epochs and between the first 5 minutes of epochs of the next day. Figure 1 describes the average absolute value of discontinuities in GPS-specific clocks and Figure 2 describes the average absolute value of discontinuities in Galileo products.  It can be seen from Figure 1 and Figure 2 that the average absolute value of discontinuities in 5 kinds of GPS-specific clock products are all less than 0.3ns for most of satellites except for some special cases, and the differences of discontinuities in 5 kinds of GPS satellite clocks are not obvious. Just as the reviewer's viewpoint, the most GPS 'discontinuities' are due to GPS clock instability over the interval of adjacent daily batches, but in essence it should be due to not only the satellite clock but also observation noise at midnight, so it is very important of the overlapping strategies for orbit and clock generation. Overall, the day-boundary discontinuities in GPS PPP time transfer with above 5 kinds of precise products are nearly at the same level and the effects can be nearly ignored. However, it can be seen from Figure 2 that the level of discontinuities in 3 kinds of Galileo clock products vary greatly, especially the discontinuities in GBM products, which even exceeds more than 6.7ns for some satellite clocks. Those high level of discontinuities in GPS and Galileo clock products provided by GFZ may be caused by the nonperfect orbit model, attitude and SRP modeling of satellites or shortage of overlapping strategies orbit at midnight because the clock products are generated thereafter orbit, and firmly consistent with orbit products, so the deficiencies in satellite orbits will directly have an impact on clocks. The high level of discontinuities in clock products may result in high level of day-boundary discontinuities in PPP station clock estimates and time comparisons due to highly dependence of PPP on externally precise products. Our experimental results in section 4.1, 4.2 and 4.3 have demonstrate these conclusions well, and the discontinuities of precise products is an significant factor for the origin of day-boundary discontinuities in PPP clock estimates. Therefore, it is challenging to employ Galileo PPP to truly reflect clock offsets of time links and employ it for long-term clock stability comparison, especially with high level discontinuities in clock products like GBM. That's the statements we would like to express in this paper.

  1. As for " Eqs 1 & 2: the j index needs to be described, too."

Thanks for the carefulness of the reviewer. Yes, we have checked the description of 'Principle of PPP' and give a detailed explanation of each variable. The variable of j (j=1, 2) in equations 1 and 2 represents carrier frequency band of satellite signal.

  1. As for " Tabs 1 & 2: should be carefully checked and corrected"

Thanks for the patience and reminding of the reviewer. Actually, we have checked the updated generating strategies that IGS and MGEX ACs used and published in related articles, posters, speeches in IGS Workshop during recent years, especially paying close attention to the updated processing strategies of orbit and clock products adopted by CODE, CNES-CLS and GFZ. We are sure of the correctness of most of the contents in Table 1& 2. But we canceled out these tables after a careful consideration and presents relating contents in the text according to References of 32-40 so as not to cause greater disagreement.

    [32] Arnold, D.; Meindl, M.; Beutler, G.; Dach, R.; Schaer, S.; Lutz, S.; Prange, L.; Sosnica, K.; Mervart, L.;    Jäggi, A. CODE’s new solar radiation pressure model for GNSS orbit determination. GPS Solut. 2015, 89(8), 775-791.

    [33] Prange, L.; Orliac, E.; Dach, R.; Arnold, Daniel.; Beutler, Gerhard.; Schaer, Stefan.; Jäggi, A. CODE’s five-system orbit and clock solution—the challenges of multi-GNSS data analysis. J. Geod. 2017,91, 345-360.

    [34] Dach, R.; Schaer, S.; Arnold, D.; et al. Activities at the CODE Analysis Center. In IGS Workshop of 2018. Wuhan, China. 29 October to 2 November, 2018.

    [35] Prange, L.; Villiger, A.; Sidorov, D. et al. Overview of CODE’s MGEX solution with the focus on Galileo, Advances in Space Research, 2020, 0273-1177.

    [36] Steigenberger, P.; Hugentobler, U.; Loyer, S.; Perosanz, F.; Prange, L.; Dach, R.; Uhlemann, M.; Gendt, G.; Montenbruck, O. Galileo orbit and clock quality of the IGS Multi-GNSS Experiment. Advances in Space Research. 2015, 55, 269-281.

    [37] Guo, F.; Li, X.; Zhang, X.; Wang, J. Assessment of precise orbit and clock products for Galileo, BeiDou, and QZSS from IGS multi-GNSS experiment (MGEX). GPS Solut. 2017, 21(1), 279–290.

    [38] Steigenberger, P.; Montenbruck, O. Consistency of MGEX Orbit and Clock Products, Engineering, 2020, 6(8), 898-903.

    [39] Loyer, S.; Mercier, F.; Capdeville, H.; Mezerette, A.; Perosanz, F. GR2 reprocessing from CNES/CLS IGS Analysis Center: specificities and results. IGS Workshop 2014, Pasadena, USA. June 23-27, 2014.

    [40] Steigenberger, P.; Fritsche, M.; Dach, R.; Schmid, R.; Montenbruck, O.; Uhlemann, M.; Prange, L.; Estimation of satellite antenna phase center offsets for Galileo, Journal of Geodesy, 2016, 90(8), 773-785.

    [41] Wang, B.; Chen, J.; Wang, B.H. Analysis of Galileo Clock Products of MGEX-ACs. The International Conference of 2019 European Navigation Conference(ENC). Warsow, Poland, 2019.

    [42] Loyer, S.; Perosanz, F.; Versini, L.; Katsigianni, G.; Mercier, F.; Mezerette, A.; CNES/CLS IGS Analysis Center: specificities and results. IGS Workshop of 2018, Wuhan, China. 29 October to 2 November, 2018.

6.As for "2.3. Network Processing Solution: needs to be described. What orbits are used? IGS or AC Final/MGEX ones? This needs to be described and full clock solutions also presented in all Figs, not just the network discontinuities."

Thank you very much for the professional comments and good suggestions. Yes, as the reviewer emphasized, we described the details of network processing solution, which can clearly viewed from line 249-273 of this paper. Actually, although it can process GNSS observations in a similar zero-difference principle as classical PPP with dual-frequency ionosphere-free combination, it requires a set of stations distributed worldwide. It can estimate the station clock offsets respect to a reference clock, unlike PPP station clock estimates respect to IGST or IGRT. The reference clock can be chosen as a station with high stability of an external time scale to ensure estimated station clocks' continuity. In this paper, Sixty ground MGEX stations distributed evenly around the world are chosen to form the set of network and seven stations with mixed multi-GNSS observations are interested. All of the interested stations, PTBB, BRUX, WAB2, SPT0, OP71, ROAG and USN8, are not only sites of MGEX but also national timing laboratories participating in computation of TAI and UTC. BRUX at Royal Observatory Belgium(Brussels) is chosen as the reference clock because of its high stability of clock. Moreover, it can be implemented with a multi-day batch to eliminate the day-boundary discontinuities at the adjacent days within one batch, and a 6-day arc we adopted in this paper.

As the reviewer's suggestions, we added the results of station clock estimates of network solution in all figures (2-16), not just the network discontinuities. However, we kept the figures of 17-22 as it is in original manuscript because our concerns are the difference of time transfer through Galileo PPP with different MGEX products, and there is no use of network solution results for comparison. As the statements abovementioned, although both of network clock solution and IPPP have advantages and are effective techniques to eliminate the day-boundary discontinuities and achieve continuous time transfer, IPPP are highly relied on products of wide-lane satellite phase bias but network solution are not. Our main focus are investigating the effect of precise GPS or Galileo products to the origin of day-boundary discontinuities in PPP station clock estimates and time comparisons, so it is enough to use network solution results for comparison to illustrate our conclusions.

  1. As for " Are Galileo-only PPP used? Some MGEX ACs (like CODE) , unlike for GPS, use a different, non standard antenna PCO/PCV atx file, this non standard Galileo PCO/PCV must be used for PPP clock solutions."

Thanks for the professional comments. Yes,compared to GPS PPP, we are more focused on PPP time transfer through new constellations like Galileo or Beidou for UTC/TAI computation. However, as our description in Section 2.2 of this paper, there are only products provided by several individual MGEX ACs and there is no combined and weighted MGEX products until now. As for the Galileo constellation, CODE, GFZ, and CNES-CLS all process Galileo observations and can provide MGEX products. However, their generating strategies of orbit and clock are very different. For example, The PCO values that CNES-CLS uses for Galileo products are conventional PCOs proposed for MGEX, while as Steigenberger, P. et.al (2016), CODE uses estimated and nonstandard PCO/PVO values for Galileo, and GFZ uses the corrected MGEX values. This will result in bias of PPP clock solutions when using above products provided by CODE, GFZ and CNES-CLS. For uniformity and better comparison, currently the PCO/PCV atx file we used are conventional MGEX PCOs/PCVs for Galileo in this paper. We definitely believe the error caused by it can be nearly ignored. However, we will carefully consider the impacts of it with ISBs in our further study.

  1. As for " Fig. 16 the plot for GBM frequency stability at 10E5 sec must be wrong, it should be worse due to the large daily discontinuities ..."

Thanks for the carefulness. With paying  close attention to the level of day-boundary discontinuities in all station clock estimates as shows in Figure 10(b)-16(b), it can be found that the amplitudes of day-boundary discontinuities in station clock estimates with COD products seems to be larger than those with GBM and GRG products at some epochs. This phenomenon can be observed in station clock estimates of PTBB at MJD 59032, SPT0 at MJD 59035-59037, and OP71 at MJD 59032 and MJD 59037. In contrast, the amplitudes of day-boundary discontinuities in station clock estimates with GRG products became higher than those with COD and GBM products at the later epochs of the study period, especially at MJD 59035-59037. The larger discontinuities at epochs of MJD 59035 to 59037 with COD and GRG products will result in worsen of frequency stability from the average time of 100000s, that is why Fig. 16 plot for GBM frequency stability at 10E5 seconds are better than that of COD and GRG. We explain it by the sentences of line 499-501, and it is consistent with Fig 10(b)-16(b) describe.

  1. As for " References: Some citations of references is not correct/appropriate, i.e. the references do not correspond to the text citations, please check."

Thanks for reviewer's carefulness, and we have revised all citations of references. New references about the strategies of ambiguity resolution, PCO/PCV that MGEX ACs of CODE, GFZ and CNES-CLS utilized for Galileo PPP or AR PPP have been added, such as Dach, R et. al (2018), Prange, L et. al (2020), Steigenberger, P et. al (2016) and Loyer, S et.al (2018).

  1. Explanation on other comments or suggestions of the reviewer " Note that COD, GRG (and likely GFZ/GBM) already routinely provide the bias information enabling AR PPP and so does IGS on an experimental bases only (see the IGS PPP Working Group) " and " With AR PPP there is little or no advantage of a network processing, besides, no network clock solutions is presented here, only the network daily discontinuities, why? "

Thanks for the professional comments and good suggestions. As stated beforehand, we described the details of network processing solution and added the results of station clock estimates of network solution in all figures (2-16), not just the network discontinuities. But we kept the figures of 17-22 as it is in original manuscript because our concerns are the difference of time transfer through Galileo PPP with different MGEX products. Although there are many advantages of AR PPP, such as accelerated convergence, improvement of the precision of station clock and frequency stability performance at 1d averaging time, AR PPP is highly relied phase bias products, and ambiguity-fixing does not improve the day-to-day continuity of satellite clocks estimated in independent daily sessions. That's no say, AR PPP is more suitable for frequency comparisons. Zero-difference network clock solution, is not highly relied on external products and can benefit from the measurements of all stations so that it is more accurate and robust than classical PPP. If a station with high stability of external time scale chosen as the reference station and its clock is estimated through PPP, then other station clock solutions are more continuous at adjacent epochs of batches. Meanwhile, AR network clock solution can be implemented in many MGEX ACs. However, it is not of all MGEX ACs could provide products for AR PPP or AR network solution needs. Moreover, it is difficult for us to apply ambiguity fixing to zero-difference network solution in a short time. To achieve our goal, it is more reasonable for us to use network clock solution with long arc batch to compare with daily PPP.

Round 2

Reviewer 3 Report

The English of the paper has been improved, but there still are possible English improvements, some, not all, are listed below.

However, as for the technical content, in my opinion, the paper needs a lot of improvements and corrections before it is acceptable for publication. More specifically:

The real GPS satellite daily clock discontinuities are small (unlike the ones shown in Fig. 1) and only the most of the Galileo ones (Fig. 2) are real and meaningful. For GPS, the Fig. 1 discontinuities are not real as they are largely due to the clock instability of GPS clocks. Perhaps, this discontinuity analysis should be either deleted, or updated/corrected, e.g., by using only the 2 30s epochs, the last of the current and 1st epoch of the next day. And acknowledging that this is subject to an uncertainty of about 0.1ns due to clock solution noise and GPS clock (in)stability. (I expect that once this is done, the conclusion will be that the GPS satellite discontinuities are small and statistically insignificant. i.e. none could be detected)

Furthermore, all the GPS PPP's should be redone with a proper weighting (1m/0.01m) and also the Galileo PPP's should be corrected and reprocessed (I expect that either a problem of the Galileo PPP implementation or an inconsistent use of Galileo products) .

Finally, if the problematic GBM Galileo clocks are kept, it should be noted that this is erroneous clock product, should not be used until corrected, and verified, GFZ should be consulted if Galileo clocks are currently corrected and problem free.

------------------detail comments/suggestions------------------

l. 40 and 41: Is this true for Galileo-only AR PPP? Any reference? I'm not sure, but it seems to me that Galileo-only AR PPP with narrow integer cycle recovery, have small daily clock discontinuity, much like for GPS-only AR PPP's. Note that GRG-CNES (from now on CNES-CLS the GRG AC designation is used) and COD MGEX products enable AR PPP for GPS and Galileo! Furthermore, Galileo HM clocks and GRG, COD integer clock solutions should improve the clock daily discontinuities.

l, 70-71: Suggest to omit this sentence: "Some researchers also believed that day-boundary discontinuities are caused by uncertainty in the estimation of phase ambiguity between .." , since it is redundant. Namely, both PPP station clock daily discontinuities and daily PPP ambiguity datum discontinuities are highly correlated and are the results of the daily average of pseudorange (colour) noise, that's why AR PPP clocks, with daily integer datum reconnection as done by BIPM, are practically continuous and free of this effect...

l. 74: what is the rinex-shift? (I've never heard it)

l. 79: daily pseudorange averages are responsible for the PPP daily discontinuities, the prove is the AR PPP!

l. 86-87: Galileo biases required for AR are readily available for GRG and COD MGEX solutions (within GRG clock files and separate bia files, respectively). In fact, the Galileo satellite biases, required for AR, are nearly constant (unlike the GPS ones) and the same ones can be used for weeks, even months. Considering this and the stable Galileo satellite PHM clocks, as well as a significantly lower Galileo pseudorange noise, the Galileo-only AR PPP timing should even now or near future outperform GPS-only AR PPP!

l. 89-90: Most ACs are using batches longer than 24h, some, e.g., CODE is 3-day data batches! Only by convention, IGS ACs must report 24h solution intervals

l. 97: Strictly speaking, ACs are not using GPS time scale, each AC time scale is realised by fixing a stable station clock, so AC time scales are not GPST! Please correct

l. 106: with different qualities (Note GRG and CODE Galileo orbit/clock solution precision (unlike for BDS ) are already compatible to GPS)

l. 122: "interesting" ? you mean typical, timing or representative ?

l. 135" " the speed of light in .."

l. 154: The weighting of 0.25m/0.06m is rather problematic, or even wrong! In fact causing a heavy PPP dependence on pseudoranges and subsequently it may result in problem discontinuities at day boundaries. A proper weighting should be about 1m/0.01 m! (note the P3 pseudorange noise typically is around 1 m for GPS and BDS and 0.5m for Galileo. The typical L3 phase noise for all GNSS is < 1 cm!

l. 171: " generated": - IGS is still generating the Final combined products

l. 199: |"zise", you mean interval ? Also should be noted that GRG and COD orbits/clocks for GPS and Galileo employ AR (ambiguity resolution) and that the bias information, necessary for AR GPS/Galileo PPP's, are readily available for GRG and COD MGEX products

l. 217 & 223 & 229: delete "kind of" or use "AC" instead

l.213-230 & Fig. 1 This DOES NOT represent discontinuities of IGS and AC GPS clocks! All it shows the GPS clock instabilities over the 5-min interval (middle of the last 5 min and the middle of the first 5-min interval of the next day). Namely, for a typical 10E-11/100s GPS frequency stability one should expect 0.3 ns difference over a 5-min interval. So all it is shown in Fig. 1 are the expected GPS clock instabilities and not the daily discontinuities! Note that the more recent GPS satellites with clock steering have much smaller differences over the 5-min intervals spanning the day boundary.

Please correct the text and Fig. 1 legend, indicating that what is shown is GPS clock instability/quality, which has no affect of GPS PPP (that's why the clocks must be solved every 30s!). Or alternatively use the last and 1st 30s interval spanning the day boundaries, acknowledging the clock solution noise and satellite clock instabilities.

l. 234-246 & Fig 2: Unlike for GPS in Fig. 1, for all the PHM Galileo satellites (Except for E11 with a Rb clock) this represent daily discontinuities, since PHM clock instability over 5-min interval are negligible (< 0.1 ns). The E26 & E30 satellites, either have PHM clock or some solution problems

Figs 1 & 2 should also include the proposed network clock solutions (it would show the same "discontinuities" (i.e., satellite clock instability, in particular for GPS!)

Please correct the text and the Fig. 2 legend accordingly. Also consider either omitting GBM clocks or changing the Fig. 2 plot range (e.g., to 0-4 ns) in order to better show COD and GRM daily discontinuities

l. 252: GBM Galileo discontinuities are wrong and should not be used to justify anything, GBM Galileo clocks should be avoided until it is fixed and corrected by GFZ. GFZ should be notified.

l. 259: PPP time scale is the one of the AC or IGS satellite clocks (see the l. 97 remark), only for IGS Final and IGR it is IGST or IGRT, resp.

l. 272: helps to reduce (there still are possible small IGS/MGEX AC orbit discontinuities, radial ones are then fully absorbed into the network clock solutions. Note the network clock and ambiguity solutions are improved (i.e. the daily clock/ambiguity datum) by averaging all the psedoranges over the day and over all the stations.

Comments on l. 253-270: How can such a network processing achieve a better clock solution results than the corresponding AC clock solutions, using more data (station), more sophisticated and consistent error modelling???

l. 280: "while seven test stations .... are chosen. All of the test stations .."

l. 290-293: Why with respect to 6-day network clocks, daily PPP station clock discontinuities can be directly compared (while taking account any (apparent) clock rates) since for the timing labs, the expected clock instabilities over the 30s day boundary should be negligible. The last and the 1st station epoch of the next day should be compared directly.

Why 6-day and not 10-day network?

l. 299: "are 622 km, .."

l. 307-310: in multi GNSS PPP (as well as global AC solutions) ISB must be solved independently from epoch to epoch.

Fig. 3-8 all PPP's should be redone with the proper weighting 1m/0.01m, (see the l. 154 comments) and the daily clock differences should be computed directly, not wrt the 6-day network solutions (see l. 290-293 remark). Using your own network solutions is akin to a circular argument. (station BRUX is missing, why?)

Fig. 9 BRUX is missing!

l. 401: ".. compared with that of GRGFinal products .." ? you mean IGS Final?

l. 406: "GFZFinal become superior to those with CODFinal and GRGFinal products, especially for station clocks of WAB2, OP71, and USN8.." ? this is contrary to Figs 4, 6 and 8 where GFZ has large daily discontinuity. Either Fig. 9 is wrong, or the Figs 4, 6 and 8 are wrong! Note the small network discontinuity, but fairly poor MDEV's at 1 day?

In fact, IGS Final and IGRRapid outperforms all other clocks, including the network one, in particular at the day MDEV! This objectively indicates that IGSFinal and IGS Rapid in fact have the smallest daily discontinuities, better than the network clocks! This should have been expected!

However, this raises another question, namely what is the MDEV performance of AR PPP clock performance with GRG or COD orbits/clocks. The answer is, still likely better than IGS Final, i.e. at 5 E-16/day see [24]. This fact needs to be clearly acknowledged here.

l. 429: why six stations, why not 7 (BRUX as well)?

l. 435: COD, GRM scale is not GPST! (GBM likely as well), please correct

l. 438: "are nearly different from the employed products in PPP." ? not sure what is meant, that station clock discontinuities do not correspond well to AC satellite clocks?

Figs 10-16 GRM and COD Galileo PPP clock solutions do not seem to be correct, they should be much better, perhaps there is some inconsistency in use (PCO/PVR? Or the use of anomalistic/eccentric E14/E18 causing the problems?)

l. 440-451: I'm not sure what is the use of this discussion. Simply said, Galileo GBM clocks are problematic, that's all what needs to be said (that that's a real problem is clear from Fig. 2! GBM should take note and look into it, correct and explain it, it should not be attempted here!)

l.451-457 and Figs 10-16 and do not look correct and indicate some problems, or inconsistency with Galileo PPP implementation/processing. That this is so it is apparent from the similar systematic errors amongst the stations and their near cancellations in the time transfers of Figs 17-20. This should be investigated and the Figs and related discussions corrected. Galileo PPP clock solutions should be about the same as GPS PPP! (Galileo orbit/clock quality for most ACs is now and was then (2019) about the same as GPS)

l. 518: "... the expected performance was not reached. "
Note: Also here, the discussion should be adjusted once Galileo PPP's are corrected. In particular, the explanation that the daily discontinuities are caused by satellite clock discontinuities, there are no observed/real significant discontinuities for GRG and COD in Fig. 2 (see above comments related to Figs 1 & 2)

l. 541-542: Why ??? There is no reason why station clock discontinuities would cancel in time transfer (it would do so only both stations are observed at the same point, using identical hardware)

l. 542-566: Is this detail discussion necessary, it is trivial and obvious (i.e. that the daily clock discontinuity in time transfer is the difference between the two station clock discontinuities, that's all what is needed here, I think)

Figs 17-20: should include the new network PPP clock solutions as well!

Also here, what was removed from each transfer? Was is the mean of all transfers, the same for all 3? What offsets was used for display purposes? All this should be clearly noted here!

Furthermore, why the same transfer was not done also for GPS and not also shown here?

Note the smaller clock noise here for the time transfers than for the individual stations (compare Figs 17-20 with Figs 10-15) and much better frequency time transfer stability than the station one (compare Fig 16 and Fig 20). This clearly demonstrates a problem with Galileo PPP implementation /processing. The time transfer noise and freq. stability should be worse by a factor up to sqrt(2), in particular for the transcontinental link!

l. 599-626 and Figs 21-23: Are these necessary? Namely, it is obvious that GBM is problematic/wrong and should not be used until it is corrected. Also note that any problematic Galileo PPP implementation (like pseudorange biases) will be reduced or eliminated in this double difference exercise. Furthermore, the stability plot of GRG-COD seems to be incorrect, it should be similar (only up to sqrt(2) larger) to the individual GRG and COD transfer stability. If this section is kept. please check and correct,.

l. 621-626: This statement does not make any sense, MGEX products can be used for time comparisons/transfers, time reference/scale of individual MGEX ACs cancels in such time comparisons/transfers.

Summary and abstract:

Should be updated, reflecting the new results, once GPS PPP is redone, with proper weighting (1m/0.01m) and the Galileo PPP's are corrected and reprocessed. Also, it should reflect that satellite daily clock discontinuities are small and that only the most of the Galileo ones in Fig. 2 are meaningful, for GPS, the discontinuities are not real as they are largely due to clock instability of GPS clock. Perhaps, this discontinuity analysis should be either deleted or updated, e.g. using only the last and 1st 30s epoch of the next day and acknowledging that this is subject of uncertainty of about 0.1ns due to clock solution noise and GPS clock stability. (I expect that once this is done, the conclusion will be that the GPS satellite discontinuities are statistically insignificant. i.e. none detected)

Finally, if GBM Galileo clock product is kept, it should be noted that this is an erroneous clock products, should not be used until corrected, and verify with GFZ it is currently corrected and problem free.

Author Response

We are grateful for the anonymous reviewer very constructive reviews of this paper. A thorough technical correction have been done as the reviewer's comments, and new results of PPP station clock estimates and frequency stability discussion for GPS (e.g. Figs 3-9) , Galileo (e.g. Figs 10-16), and for time comparisons of GPS & Galileo (Figs 17-24) have been redone and reprocessed, with proper weighting of 0.3m/0.003m for GPS and 0.22m/0.004m for Galileo according to the code and phase noise. From the results, it indeed can be clearly seen that most of GPS & Galileo satellite daily clock discontinuities are small, and not particularly critical to time comparison. Moreover, we corrected the GPS clock daily discontinuities using only the last and 1st 30s epoch of the next day, and adjusted the statements. The combined GPS 'Final' and 'Rapid' clock discontinuities are subject of uncertainty of nearly 0.1ns due to clock solution noise and clock instability. However, the daily discontinuities in CODFinal, GFZFinal and GRGFinal products are previously larger, and the quantities are nearly the same and at the level of 0.25ns for most of GPS clocks. Meanwhile, the related text in 'Abstract' and Summary' has been adjusted according the corrected GPS & Galileo PPP clock estimates and time comparison results.
